**Impact Webs: A novel conceptual modelling approach for characterising and assessing complex risks**

**Authors:** Edward Sparkes[1], Davide Cotti[1], Angel Valdiviezo Ajila [2], Saskia E. Werners[1, 3], Michael Hagenlocher[1]

1) United Nations University - Institute for Environment and Human Security, Germany
2) Escuela Superior Politécnica del Litoral, Ecuador
3) Wageningen University & Research, the Netherlands

Correspondence to: Edward Sparkes (esparkes1@gmail.com)

**Abstract**

Identifying, characterising and assessing the complex nature of risks is vital to realise the expected outcome of the Sendai Framework for Disaster Risk Reduction. Over the past two decades, the conceptualization of risk has evolved from a hazard-centric perspective to one that integrates dynamic interactions between hazards, exposure, systems vulnerabilities and responses. This calls for a need to develop tools and methodologies that can account for such complexity in risk assessments. However, existing risk assessment approaches are hitting limits to tackle such complexity. To this aim, we developed a novel complex risk assessment methodology named 'Impact Webs', inspired by a conceptual risk modelling approach named Climate Impact Chains that integrates aspects of various other conceptual models used in risk assessments such as Causal Loop Diagrams and Fuzzy Cognitive Mapping. Impact Webs are developed in a participatory manner with stakeholders and characterise and map interconnections between risks, their underlying hazards, risk drivers, root causes, responses to risks, as well as direct and cascading impacts across multiple systems and at various scales. In this methodological paper, we show how we developed the Impact Web methodology, including how we derived which elements to include in the model, demonstrating the logic and visual output and listing the steps we followed for construction. As proof of concept, we present the results of a complex risk assessments in Guayaquil, Ecuador, which investigated how COVID-19, concurrent hazards and responses propagate risks and impacts across sectors and systems during the pandemic. Reflecting on the utility of Impact Webs, application in case studies demonstrates the methodologies' usefulness for understanding complex cause-effect relationships and informing decision-making across different scales. The participatory process of developing Impact Webs with stakeholders uncovers critical elements in systems at risk, and helps to evaluate co-benefits and trade-offs of decisions by uncovering how the outcomes of disaster risk management practices affect people, organisations and sectors differently. Offering a system-wide perspective for modelling, Impact Webs stand as a valuable methodological contribution for complex risk assessment.

**Copyright statement:**

## 1. Introduction

Identifying, characterising and assessing the complexity of risks is vital to realise the expected outcome of the Sendai Framework for Disaster Risk Reduction (UNDRR, 2022). As sectors and systems become increasingly interconnected, the space in which risks can cascade is expanding (Helbing, 2013; UNDRR, 2022). This has been starkly evident throughout the COVID-19 pandemic, where impacts have not just arisen in the health system, generated by the hazard, but also from the cascading effects of impacts and from societal responses through global lockdowns, with different regions suffering from vastly different consequences depending on underlying societal vulnerabilities and the resilience of their systems (Hagenlocher et al., 2022). These characteristics are not limited to COVID-19, and have also been observed in other contexts, including from the compounding and cross-border effects of extreme

climate events (Simpson et al., 2021; Zscheischler et al., 2018), or from the global ripple effects of
armed conflicts (Cui et al., 2023).
Over the past two decades, the conceptualization of risk has evolved from a hazard-centric perspective
to a more encompassing notion that integrates the dynamic interactions between hazards, exposure,
vulnerability (IPCC, 2014) and, more recently, response risks (i.e. risks that can arise from responses
to risks and impacts) (Simpson et al., 2021; Ara Begum et al., 2022; IPCC, 2023; Hagenlocher et al.,
2023). Different terminologies have been used to conceptualise these dynamic interactions, including
cascading, compound, and systemic risks. In this paper we use the term 'complex risks' to encapsulate
these different risk framings. Given that complexity is now understood as a defining feature of risks,
single-hazard and single-risk approaches, while useful in certain contexts, are becoming increasingly
insufficient for comprehensive disaster risk management (Simpson et al., 2021; UNDRR, 2022;
Schlumberger, et al., 2024; Sett et al., 2024; de Ruiter & van Loon, 2022). This has been recognised
by the Intergovernmental Panel on Climate Change (IPCC) in the Sixth Assessment Report, which
notes that risks and responses, including their determinants, can all interact dynamically in shaping the
complexity of climate risk (Ara Begum et al., 2022). Additionally, the Global Assessment Report 2022
(GAR 2022) from UNDRR stresses the importance of understanding and assessing the complex nature
of risks as a key foundation for risk informed decision making (UNDRR, 2022). However, existing data
driven and quantitative modelling approaches are hitting limits to tackle such complexity. The combined
effects of multiple hazards, threats or shocks should not be assessed just through the addition of each
of their impacts independently, but instead require systems approaches to understand risk and impacts
(de Ruiter et al., 2020; Ara Begum et al., 2022; Hagenlocher et al., 2023; de Brito et al., 2024). There
is therefore a need to develop methodologies that take a system-wide lens for analysis, that can account
for how multiple hazards and vulnerabilities of systems and sectors interact to better understand
complex risks.
To this aim, we developed a novel complex risk assessment methodology named 'Impact Webs'. Impact
Webs are inspired by a conceptual risk modelling approach named Climate Impact Chains (see Menk
et al., 2022 for a review of applications), and draw inspiration from various other conceptual models
used in risk assessments. Climate Impact Chains were originally developed for sectoral climate risk
assessment (Schneiderbauer et al., 2013; Zebisch et al., 2023, 2021; Hagenlocher et al., 2018), in
which elements of the model are assigned to the key risk components used in disaster and climate risk
assessments of hazard, exposure and vulnerability, and cascading effects are assigned as intermediate
impacts. One critique of Climate Impact Chains is that they often depict a linear cause-effect relationship
for a single sector or hazard, and thus do not capture the complexity of systems interaction well (Harris
et al., 2022). With Impact Webs, we built on Climate Impact Chains, integrating aspects of system
mapping approaches such as Causal Loop Diagrams (e.g Coletta et al., 2024; Groundstroem & Juhola,
2021; Dianat et al., 2020; Rehman et al., 2019), Fuzzy Cognitive Maps (e.g. Gómez Martín et al., 2020;
Ahmeda et al., 2018; Chandra & Gaganis, 2016) and Bayesian Belief Networks (e.g. Malekmohammadi
et al., 2023; Scrieciu et al., 2021; Bashari et al., 2016; Giordano et al., 2013). With this, we aimed to
integrate the key risk components in disaster and climate risk assessments with a systems perspective
to identify, characterise and map interconnections between risks, their underlying hazards, risk drivers,
root causes, responses to risks, as well as direct and cascading impacts across multiple systems and
at various scales. Impact Webs aim to better account for the complexity of risk interaction compared to
Climate Impact Chains, by developing flexible and less linear conceptual models that can help to
understand complex risks.
In this paper, we offer a new complex risk assessment methodology in the form of Impact Webs,
detailing how we developed it. To do this, we first conducted a scoping review of literature on conceptual
risk models that we drew inspiration from. Informed by the review, we identified constitutive elements
for the model and developed a graphical structure. We then developed key steps for doing a complex
risk assessment with Impact Webs, testing our methodology in five cases. These cases were Coxes
Bazar humanitarian camp (Bangladesh), the Sundarbans region (India), a national scale assessment
(Indonesia), the Maritime region (Togo) and the city of Guayaquil (Ecuador).  The complex risk
assessments investigated how COVID-19, concurrent hazards (e.g. hydrological, geophysical,
climatological) and responses too them (e.g. restriction measures) interacted with underlying societal
vulnerabilities to propagate risks and impacts across sectors and systems during the pandemic
(Hagenlocher et al., 2022). COVID-19 was selected as the entry point for the risk assessments as the
pandemic has been so diverse and cross-scale in its effects, therefore such an event was ideal to test
a novel risk modelling approach for understanding complex risks. As proof of concept, we present the
results and final output from one of the five test cases, showing an Impact Web and narrative storyline
for the city of Guayaquil, Ecuador during the COVID-19 pandemic. Guayaquil was selected to
demonstrate our proof of concept due to the city's high vulnerability and exposure to the compounding
effects of multiple hazards and the presence of many drivers of risks creating numerous challenges for
risk management, therefore making it a fitting case to showcase a new risk assessment methodology.
The remainder of the paper is structured as follows: In section 2, we present the methodology for
developing Impact Webs, which includes the scoping literature review of conceptual risk models, the
constitutive elements we selected to populate the model, and the steps that were followed during the
complex risk assessments to construct an Impact Web. In the results in section 3, we show our proof
of concept, presenting the Guayaquil test case. In the discussion in section 4, we reflect on the utility of
Impact Webs, looking at strengths, limitations and potential future research directions. We conclude in
section 5 with synthesis of the paper, highlighting Impact Webs as a conceptual model that moves
beyond single-risk or single-hazard assessment, which can be used as an approach for system-wide
complex risk assessment.

## 2.      Methodology

In section 2, we present our methodology to develop Impact Webs. We show our methodological pre-
development, with a scoping review of other conceptual risk modelling approaches we drew inspiration
from. We then elaborate on the elements that were selected in the model, introduce the five test cases
and present the steps we followed to construct an Impact Web.
### 2.1.    Methodological pre-development: Scoping review of conceptual risk models for inspiration

Given that we aimed to develop an approach that took a systems perspective for analysis to better
understand complex risks, we conducted a scoping review of literature on conceptual risk models which
do this. The scoping review was non-systematic and not meant to be exhaustive. It was done to support
methodological synthesis and inspire the concept development for our approach by looking at features
of different methodologies that could be useful. A non-systematic scoping review was chosen it this
type of review approach has advantages for developing new methodologies. Non-systematic scoping
reviews allow for exploratory flexibility, drawing on grey literature, emerging studies and integration of
methodological aspects that authors had used in past research. This supported creative synthesis by
combining ideas from various disciplines (Munn et al., 2022). Texts were selected and reviewed based
on authors own experience, expert judgement, and searching using the Scopus search engine.   A
general description of the approach's features is given, as well as the strengths and weaknesses in a
complex risk context. We also provide selected key references that inspired us (see Table 1).
**Table 1:** Overview of conceptual models used in risk assessments

| Approach | Features | Strengths in a complex risk context | Weaknesses in a complex risk context | Key references |
|---|---|---|---|---|
| Climate Impact Chains | *Model illustrates key risks and their drivers for a specific context, with elements assigned to* | *Opportunities pertain to the flexible and relatively simplistic form, making them more easy to* | *Analytic emphasis on linear cause–effect relationships, neglecting and oversimplifying* | *Sett et al (2024)*<br><br>*Petutschni* |

| | | | | |
|---|---|---|---|---|
| | *hazard, exposure, vulnerability and intermediate impacts recognising that the system is affected by multiple risks that need to be prioritised* | *develop through a participatory process, allowing for perspectives of vulnerable groups and impact dynamics for specific case studies.*<br><br>*Innovative focus on intermediate impacts, making them conducive to analyse cascading impacts, as well as focus on risk drivers and the "cause–effect relationships" that define them*<br><br>*Can identify entry points for adaptation across the model elements, including for risk drivers and root causes* | *complex system interactions*<br><br>*Narrow definitions of system boundaries*<br><br>*Limited applicability to fragmented governance landscapes (in consideration of risk ownership), resulting in 'blind spots' for adaptation and response risks*<br><br>*Given the often strong participatory focus, expert facilitation is needed. This can be useful for managing power dynamics during the modelling exercise and for explaining conceptualisations of risk to non-experts* | *g et al (2023)*<br><br>*Zebisch et al (2023) Harris et al (2022)*<br><br>*Hagenloch er et al., (2018)* |
| Fuzzy Cognitive Maps | *Semi-quantitative diagramming tool that maps the important elements of a system in nodes, providing the relationship between nodes in terms of direction and strength* | *Indicate the strength of the causal relationships (weak, medium, strong) and the ability to examine feedback effects in systems where exact relations are hard to quantify*<br><br>*The vector-matrix structure facilitates the aggregation of different stakeholders' views, which is affective for participatory modelling exercises*<br><br>*Can integrate temporal considerations by introducing delays in the model assuming that the weights can change over time, which is useful for assessment of the delayed cause-effect of relationships* | *Risks force-fitting archetype to the systems problems, rather than as a lens to look at the system from different perspectives*<br><br>*Results can be difficult to communication to non-experts*<br><br>*Often a lack of analysis on the difference in perspectives between stakeholders, leading to analysis that accounts for the trade-offs among co-benefits of interventions, and not for trade-offs between stakeholder's valuations* | *Scrieciu et al (2021)*<br><br>*Gómez Martín et al (2020)*<br><br>*Ahmeda et al (2018)*<br><br>*Chandra & Gaganis (2016)* |
| Causal Loop Diagrams | *Tool for visualising the causal structure and delays between interacting system elements, demonstrating how change in one variable can influence others by reinforcing or balancing them, helping to describe how complex interconnections and feedback loops affect the* | *Provide insights into behavioural trends and stakeholders interactions affected by risks as well as response measures, making them useful to support decision-making processes at a planning/ strategic level*<br><br>*Allows for an examination of potential* | *Inadequate representation of spatial dynamics*<br><br>*The isolation and examination of specific dynamics may produce results which are misrepresentative of the system functioning as a complex whole* | *Hanf et al (2025)*<br><br>*Coletta et al (2024)*<br><br>*Groundstro em & Juhola (2021)*<br><br>*Dianat et al* |

| | | | | |
|---|---|---|---|---|
| | *systems dynamic evolution* | *future trajectories of change based on whether feedback loops are reinforcing (indicating a dynamic situation) or balancing (indicating a more stable situation)*<br><br>*Often conducted in a participatory manner, obtaining data coming from formal and non-formal sources*<br><br>*Can be combined with quantitative indicators to create 'what if' scenarios that project how changes in one indicator (for example, by implementing a response measures) can make changes in other parts of the system* | *Difficult to validate robustly, particularly affecting reliability when assessing social, economic and political sub-systems, which are more difficult to predict than physical based sub-systems*<br><br>*Does not distinguish between physical and information links* | *(2020) Rehman et al (2019)* |
| Influence Diagrams | *System elements connected by arrows, indicating causal links through symbols that make distinctions between stocks & flows of information & physical assets, often to model a decision-making process* | *Making distinctions between stocks & flows of information & physical assets forces the modeller to think about operational factors of the model early in the modelling process*<br><br>*Excel in identifying the effects of interventions in response to risks across different social-ecological systems*<br><br>*Through stakeholder input, they can represent the socially constructed nature of risks, and therefore can identify groups or individuals who perceive more system relationships and risks and thus have more insight into how to change the system* | *The greater level of detail requires many conventions and rules, which may not be easy to communicate to non-expert stakeholders*<br><br>*Defining and assessing variables and strength of links can be seen as an exercise in power, in which dominant bodies can more strongly influence decision variables and 'push' the system into their preferred direction* | *Malekmoha mmadi et al (2023)*<br><br>*Mühlhofer et al (2023)*<br><br>*Parviainen et al (2019)*<br><br>*ElSawah et al (2015)* |
| Bayesian Belief Networks | *Integrate qualitative data in the form of cause and effect diagrams and quantitative data in the form of assigning a value to the strength of the dependence between variables using conditional probability, offering a probabilistic representation of the relationships between system elements and* | *They can be used to perform sensitivity and scenario analysis, thereby allowing decision makers to predict the more probable outcomes of interventions in response to risks and identify management actions that are most likely to lead to specific outcomes* | *Use directed acyclic graphs which cannot contain cycles or feedback loops*<br><br>*A large amount of data is required for populating the conditional probability tables, which is a challenge in data scarce contexts*<br><br>*A long cause-effect* | *Malekmoha mmadi et al (2023)*<br><br>*Scrieciu et al (2021)*<br><br>*Bashari et al (2016) Giordano et al (2013)* |

| | | | |
|---|---|---|---|
| | *how they influence one another* | *The conditional probability tables used with the cause and effect diagrams can be updated when new data generated or collected, for example from climate models, case studies or monitoring programs*<br><br>*Link well with other conceptual modelling approaches to model quantitatively and assess uncertainty* | *chain of nodes can show reduced sensitivity, which can propagate uncertainty from parent nodes to child nodes. This incentivises reducing the models complexity, which does not reflect risk in complex systems* | |


### Lessons from the review


Different conceptual modelling methodologies have been applied across disciplines for assessing
complex risks which have provided useful lessons for our approach. From the papers we reviewed,
Influence Diagrams and Bayesian Belief Networks show usefulness to understand interactions of
biophysical processes such as extreme events, with additional dynamic inputs, such as interventions in
response to risks or stakeholders' perceptions of risks and risk management decisions (e.g. Scrieciu et
al., 2021). Causal Loop Diagrams and Fuzzy Cognitive Maps provide a useful framework to examine
interconnections and feedback effects between elements in one or multiple systems to support
integrated and cross sectoral decision making (e.g. Hanf et al., 2025; Dianat et al., 2021), and Climate
Impact Chains are effective for eliciting stakeholder knowledge due to their flexible and relatively
simplistic form, which is useful to develop shared system understanding and co-create policy
recommendations, and their innovative focus on intermediate impacts makes them conducive to
analyse cascading impacts (e.g. Sett et al., 2024). It is important acknowledge that the approaches in
Table 1 are not mutually exclusive, and do cross over with one another. Methodological combinations
of approaches are common and adjusted to suit the decision context or setting of the risk assessment.
For example, there is often integration between Fuzzy Cognitive Maps, Influence Diagrams and
Bayesian Belief Networks.

With Impact Webs we drew on observed strengths in the literature we reviewed, aiming to create a
model that is useful for; 1) understanding interactions of impacts from extreme events, stakeholders'
responses to them, as well as stakeholders perceptions of risks and risk management, 2) examining
interconnections and feedback effects in one or multiple systems to support integrated and cross
sectoral decision making, and 3) eliciting stakeholder knowledge to develop shared system
understanding and co-create policy recommendations. In order to achieve these aims, with Impact
Webs we built on the hazard, exposure, vulnerability framing from Climate Impact Chains, which is
useful to understand how risks emerge from extreme events (e.g. Hagenlocher at al., 2018; Sett et al.,
2024), expanding this to include aspects such as feedbacks and non-linear interconnections which are
well suited to Fuzzy Cognitive Maps and Causal Loop Diagrams (e.g. Hanf et al., 2025; Coletta et al.,
2024; Ahmed et al., 2018). To do this we included the dynamic interaction of multiple hazards, threats
and shocks, multiple exposed elements and the impacts to exposed elements. We additionally included
the drivers and root causes of vulnerabilities to exposed elements in Impact Webs. Including drivers
and root causes in the model helped us to understand not just what impacts occurred, but also why
they occurred (Blaikie et al., 1994; Wisner et al., 2004). Drawing on strengths of Influence Diagrams
and Bayesian Belief Networks (e.g. Mühlhofer., 2024; Scrieciu et al., 2021), we included interventions
in response to risks and impacts, as well as response risks arising from them. We did this as it was
important for us to align with the most recent IPCC risk framing (Simpson et al., 2021; Ara Begum et
al., 2022). All types of approaches we reviewed use graphical methods to show cause-effect
relationships, most commonly using arrows and symbols to signal a relationship and influence. We also

adopted this, using graphical methods to show cause-effect relationships and feedbacks. The majority of studies we reviewed integrate some form of input from stakeholders. However, it was common for stakeholder participation to decrease with increasing complexity of the method used, due to difficulties in communicating and facilitating the approach (Parviainen et al., 2019). This was an important lesson for us from the review. We aimed for a strong participatory approach that involved collaboration and integration of different expertise and knowledge. Given this, drawing on the strengths of Climate Impact Chains (Harris et al., 2022) we aimed to make the steps for developing the model simplistic so that stakeholders were not overwhelmed and could be highly engaged during the modelling process. This helped to identify key system elements that stakeholders felt were important, valued highly and wanted to protect from risks and impacts, for example key economic sectors. A systems thinking perspective was commonly taken towards analysis in all approaches. We did the same for Impact Webs. The aim of taking a systems perspective was to enhance systems' understanding and reduce uncertainty through modelling non-linear interactions and dynamics. We also wanted to model interactions across different scales (i.e. from global to local). Therefore, we expanded beyond a sectoral focus (e.g. drought risk for the agriculture sector) often used with Climate Impact Chains, aiming to capture cross-sectoral risks, impacts and vulnerabilities and their influences between one another. Lastly, we observed in the majority of papers that the visual output of the model was also accompanied by narrative-based methods used to explore and communicate findings (e.g. Hanf et al., 2025). We followed this by including a narrative storyline which described the findings of the assessment and described the Impact Web in a structured and relatable way.

## 2.2. Selection of constitutive elements in an Impact Web

Building on the lessons from the scoping review, here we present the elements that were selected for visualisation in an Impact Web. We elaborate on why these elements were selected, including the conceptual backing for choosing them and system interactions we wanted to assess.

### Hazards, threats & shocks

Conceptual risk models are developed to better understand impacts arising from a hazard, threat or shock, such as hydrological extremes (e.g. flood and drought), biological hazards (i.e. COVID-19 or a cholera outbreak) or geopolitical aggression (e.g. a war or conflict). We wanted our model to improve understanding of compounding interaction, given the increasingly interconnected nature of multi-hazards impacts on sectors and systems (UNDRR, 2022). Therefore, we included multiple hazards, threats and shocks to the system being modelled in Impact Webs.

### Impacts

Impacts were the second element we included after our scoping review (Sett et al., 2024; Zebisch et al., 2023; Lawrence et al., 2020). This was done to identify direct negative impacts from hazards, threats and shocks, as well as cascading impacts and any potential positive impacts that may have arisen (often as a result of interventions) in the system being modelled. Modelling cascading impacts, which arise through impact propagation (Mühlhofer et al., 2023; Carter et al., 2021), helped to understand the systems interconnectedness as linkages between sectors and sub-systems could emerge as connections were characterised. In the visualisation of the model (See Figure 1) we do not make a visual distinction between direct and cascading impacts. There is however, a conceptual distinction as every impact that is not directly connected to a hazard threat or shock can be understood as cascading. Additionally, through modelling impacts, the compounding effects of multiple hazards, threats or shocks occurring simultaneously could be analysed (Simpson et al., 2023). We did make a visual distinction between negative and positive impacts, using a cross and a tick (see Table 2).

### Interventions

With Impact Webs, we wanted to characterise and assess how decisions in response to or anticipation of risks and impacts have impacts in systems. Drawing on aspects of Fuzzy Cognitive Mapping, Influence Diagrams and Bayesian Belief Networks, which are useful for modelling the effects of

decision-making processes (Scrieciu et al., 2021), as well as the more recent framing of response risks
(Simpson et al., 2021; IPCC, 2023; Hagenlocher et al., 2023), interventions were included in our
conceptual model. When developing the model, both positive and negative impacts from interventions
were included (e.g. the negative impacts that occurred because of COVID-19 lockdowns). The defined
decision context and system boundaries denote the granularity of response risks and impacts included
in the model, for example whether mapping the city level or intergovernmental level interventions.
**Risks that did not manifest**
When modelling a system that has been affected by hazards, threats or shocks, there can be potential
adverse consequences that are avoided, often as a result of interventions (e.g. the risk of a healthcare
system collapsing, or a bread basket failure). We included these in Impact Webs and named the
element risks that did not manifest. These are conceptually different from positive impacts, as they are
potential negative consequences that did not happen.
**Drivers of risk**
Understanding causality is a key rationale for disaster risk assessment (Oliver-Smith et al., 2017), and
taking a systems approach facilitates looking into causal connections that can deepen the assessors
understanding of how and why impacts can emerge (Gómez Martín et al., 2020; Coletta et al., 2024).
Therefore, an important element for our model was to look at what was drivers of risks and impacts in
the system. Drivers of risks are processes or conditions that influence the level of risks and impacts by
increasing levels of exposure and vulnerability, or reducing the capacity of people to manage or adapt
to risks. We were motivated to include drivers as it asks the modeller to critically reflect on how and
why societal functions, essential sectors, system elements or stakeholders were adversely affected due
to high susceptibility or low coping/ adaptive capacity.
**Root causes of risk & vulnerability**
An additional step to further understand causality was to model root causes of risk and vulnerability.
These are underlying factors that influence drivers of risk (Blaikie et al., 1994; Wisner et al., 2004;
Zebisch et al., 2023). Including them supported exploring socio-economic and political structures and
processes and choices that further explain why a particular community, sector, system or place is at
risk in the first place, and are important for designing risk management to be sustainable and lasting.
Both drivers of risks and root causes are often distant spatially and temporally from the system under
investigation (Wisner et al., 2004), they are however, highly relevant when wanting to understand
complex risks.
**Connections between elements**
Following the other conceptual modelling approaches we reviewed, we used graphical methods to show
connection between our chosen elements and visualise risks. We selected arrows to indicate directional
cause-effect relationships. Given the limitations of directed acyclic graphs used in many Bayesian Belief
Networks and Influence Diagrams in showing feedback effects (Bashari et al., 2016), we took an
approach more inspired by Causal Loop Diagrams. This meant we could better demonstrate indirect
effects and feedback loops (Groundstroem & Juhola., 2021), which is both more appropriate to a
complex risks context and helped us understand interconnectivity between elements. We used different
colours and dashed arrows to show connections coming from drivers of risk and root causes of risk &
vulnerability, compared with using black arrows for other elements in the model (see Table 2). This was
done so a quick and engaging visual distinction could be made for external stakeholders working with/
viewing the model.
**Scales**
From our review we did not find conceptual modelling approaches that were effective at demonstrating
risk elements and their interactions across spatial scales. For example, a critique of Impact Chains and
Fuzzy Cognitive mapping approaches is often that they have narrow definitions of system boundaries
(Petutschnig et al., 2023; Ahmed et al., 2018). For Impact Webs, we included three spatial scales in our
model (i.e. local, regional and global), which was intended to model globally networked risks, as well as
demonstrate risk drivers, root causes and impacts that are often spatially distant but have effects in the
local context (Helbing, 2013). As the test case study contexts where we made Impact Webs were
geographically diverse (see Section 2.4), there was flexibility in how the 'local' scale boundary was
defined. For example, for the Coxes Bazar case, the local scale was defined to inside the humanitarian
camp. Comparatively, Guayaquil focused on investigating the city municipality, whereas Indonesia was
at the national scale.

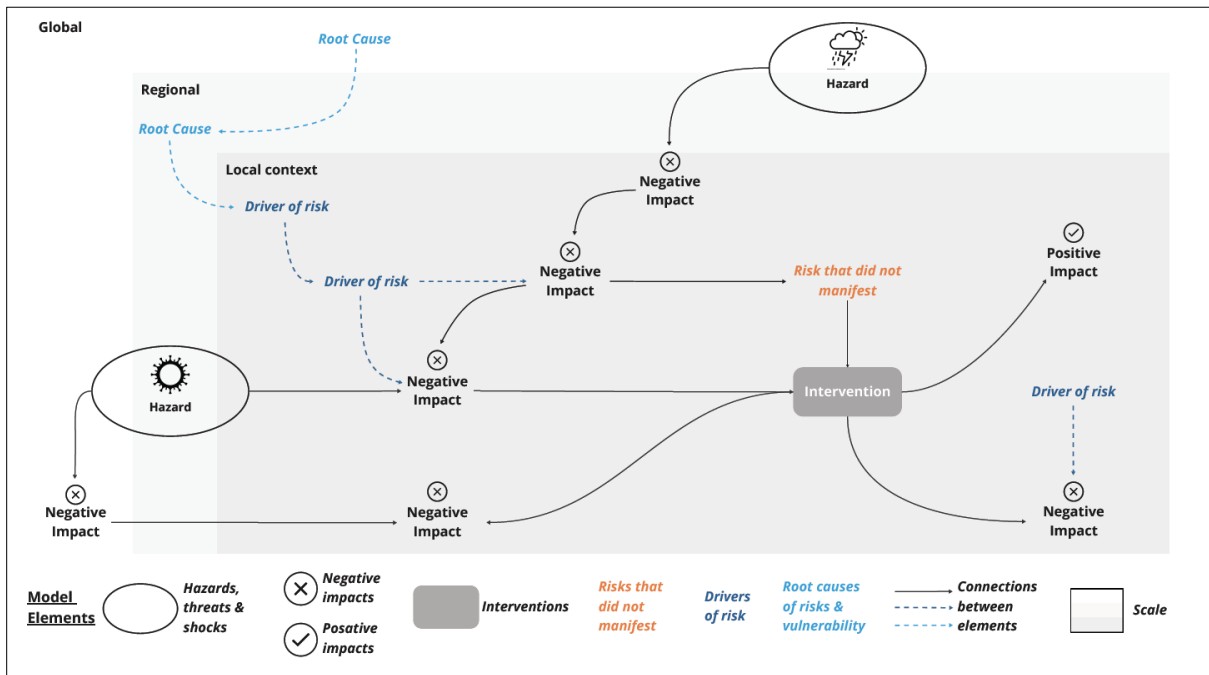

**Figure 1:** *Elements and possible graphical structure of an Impact Web. While here we present our*
*chosen graphical output of the conceptual model with computerised tools, an Impact Web could equally*
*be made with a pen and paper, for example if being developed in a community workshop. The model*
*maps the direct and cascading impacts and their interactions resulting from a biological and climate-*
*driven hazard. These impacts trigger an intervention, which results in further negative and positive*
*impacts (i.e. response risks), as well as a risk that did not manifest. Drivers of risk and root causes*
*linked to why impacts emerge are also included. The elements in the model are predominantly focused*
*on the local context, however, important regional and global interactions are included.*
**Table 2:** Description of the elements used in Impact Webs, including the chosen visual representation
in the model and examples

| Element | Description | Visual representation in the model | Examples |
|---|---|---|---|
| Hazards, threats & shocks | A potentially damaging, sometimes unknown natural or human-made phenomenon, event or activity that may have adverse effects on vulnerable and exposed elements in a system. They may emerge slowly (e.g. sea level rise, droughts) or rapidly (e.g. flash flooding, earthquakes) and can be from a single phenomenon, event or activity (e.g. cyclone) or multiple interacting phenomena, events or activities (e.g. cyclone occurring during | (Icons used to visually communicate hazard, threat or shock) | COVID-19, tropical cyclone, flood, drought, armed conflict |

| | | | |
|---|---|---|---|
| | a pandemic; compounding heat and drought) | | |
| Negative impacts | Negative effects caused by one or multiple hazards, threats, shocks or policy responses to them. They can direct or indirect (i.e. cascading). A cross is placed next to the text in the model to visually communicate the impact is negative. | ⊗ | Increased mortality, loss of income, disruption in remittances, reduction in agricultural production, mental health effects |
| Positive impacts | Positive effects that occur within the system. They are often caused by interventions in response to hazards, threats or shocks. They can direct or indirect (i.e. cascading). A tick is placed next to the text in the model to visually communicate the impact is negative. | ⊘ | Advances in digitalization, improved early warning systems, increased risk awareness |
| Interventions | Actions that are taken in response to an impact, risk or hazard, threat or shock | | Lockdown during a pandemic; provision of food and shelter items during a tropical cyclone; trade embargo with a country due to armed conflict; construction of a water reservoir to deal with droughts |
| Risks that did not manifest | Potential adverse consequences for human, technological and/or ecological systems that did not occur, often because of interventions in response to hazards, threats or shocks | *Risk that did not manifest* | Collapse of the health-care system during a pandemic, crop failure leading to food system collapse, fuel shortage crisis, economic recession |
| Drivers of risks | Processes or conditions that influence the level of risk by increasing levels of hazard, exposure and vulnerability | *Driver of risk* | Poverty, lack of functioning early warning systems, large informal work sector |
| Root causes of risks & vulnerability | Underlying factors influencing drivers of risk. They may be geographically or temporally remote as they often stem from structural as well as social, economic, cultural and political conditions that are difficult to influence directly | *Root cause of risk & vulnerability* | Regional development challenges, endemic corruption, colonial legacies |
| Connections between elements | Connecting arrow that indicates directional cause-effect relationships between elements in the system. The light blue dashed arrow is used to illustrate connections from a root causes of risk & vulnerability. The dark blue dashed arrow is used to illustrate connections from drivers of risk. The back arrow is used for all other connections. Different colours are used to make a visual distinction between the influence of drivers of risk and root causes of risk & vulnerability and other elements in the model | →  - - - →  - - - → | An increase in COVID-19 cases leading to a |
| Scale | The geographic scale at which the system interaction is occurring | | Local, municipal, river catchment, county, national, regional, continental, global (in |

| | | | Figure 1, we show local context, regional and global as an example) |
|---|---|---|---|
| | | | |


### 2.3.   Trail in test cases

Impact Webs were developed in five test cases to assess complex risks (Hagenlocher et al., 2022).
This was done to trail our methodology with groups of stakeholders across diverse case study contexts.
Doing this had three purposes. First, it allowed for adjustment in the steps for construction (see section
2.4) and improvement of the methodology through stakeholder feedback. Second, we could test Impact
Webs across different locations each with their own unique challenges and characteristics, building
from the same entry point to see if the approach was replicable and a useful risk assessment tool in
different contexts. Third, we wanted to develop a methodology that was participatory, therefore we
needed to trail it with stakeholders to learn how they would engage with developing such a model. We
trailed the methodology in the cases between June and September 2021, using COVID-19 as the entry
'seed' element, building from there and adding additional elements to populate the model using desk
study and stakeholder workshops. COVID-19 was selected as the first hazard to start build the model
around as the pandemic had been a situation that challenged conventional risk and hazard settings,
and was therefore a unique event in which to test a new complex risk assessment methodology. The
cases were chosen to cover a wide thematic range. In this paper we only present the final Impact Web
for one of the five cases (Guayaquil, Ecuador), to demonstrate our proof of concept (see section 3).
The test cases were as follows:

• Coxes Bazar humanitarian camp (Bangladesh): Showcased COVID-19 and pre-existing
social inequity in a challenging and fragile setting. The case highlighted characteristics of
vulnerable people and communities living in highly dependent systems.

• Sundarbans region (India): Encompassed a strong multi-hazard perspective, demonstrating
the concurrence of COVID-19 with tropical cyclone Amphan. The case exhibited the dynamic
nature of complex risks by exploring the delay between the cause and effect of impacts.

• National scale (Indonesia): Highlighted how COVID-19 and other hazards led to
interconnected challenges on all fronts. The case had a special focus on the role of social
protection.

• Maritime region (Togo): Focused on the rural-urban and national-international interlinkages of
systems and how they were affected by COVID-19 and concurrent hazards in a regional Sub-
Saharan context with high levels of poverty.

• Guayaquil (Ecuador): Gave specific insights into how COVID-19 and other hazards
overwhelmed a densely populated, overcrowded urban setting. The case presented
characteristics of tipping points and how system dependencies from global to local scales
created and reinforced vulnerabilities (see section 3 for more).


### 2.4.   Steps for constructing an Impact Web

Here we present the steps that we followed to construct an Impact Web in the test cases, which was
informed by lessons from the scoping review and from previous experiences among the research
team of undertaking risk assessments.

#### Step 1: Scoping

Risk assessments are done in a specific setting to support decision making processes. Following in the
steps of risk assessments that have been successful in the past (e.g. Zebisch et al., 2023; Hagenlocher
et al., 2018), the preliminary step for constructing an Impact Web was the scoping. Here, we defined

objectives and the need for the multi-hazard risk assessments across each case, considering how the conceptual models could enhance understanding and inform decision making that reduced risks. While systems theory denotes that system boundaries change, for example due to shifting climatic conditions (Steffen et al., 2015), practically, selecting the scale to model across the test cases helped to refine decision context. This was done through looking at geographical or administrative boundaries to select the area of primary focus. We then identified critical societal functions, essential sectors and key elements at risk in each of the cases, as well as key stakeholders that were engaged later in the process. Once this was defined however, it was important that there was flexibility when populating the Impact Web with elements, given that we wanted to model cross scale dynamics including feedback effects, cascading effects and globally networked risks that were identified outside the geographic boundaries of the test cases (Helbing, 2013; Sparkes & Werners., 2023).

### *Step 2: Identifying and mapping a preliminary number of elements*

While there are not restrictions in terms of the order for selecting the elements in an Impact Web, we found it was preferable to start from a limited number of key elements that you want to better understand and then progressively build up the causal connections. In our test cases, we wanted to understand multi-hazard interaction of COVID-19 and concurrent hazards, threats and shocks, therefore COVID-19 was the logical entry point. This perspective acknowledged that the systems complex relationships emerge more clearly when under stress, i.e. when direct and cascading impacts occur, the connections between them and hazards become more visible and therefore easier to observe. In this sense, the first number of elements functioned as "seeds" for the identification of the systems interdependencies. We found building from key hazards, threats and shocks as the 'seed' elements facilitated following a more simplistic cause-effect chain at the start of construction, i.e., direct impacts arising from each of the hazards, threats or shocks. From direct impacts, cascading impacts then interventions and response risks, and finally drivers of risks and root causes followed. While Impact Webs eventually aim to map risk complexity, we found it difficult to start from the more complex interactions (i.e. feedback effects). Rather, starting with more simple connections is easier for the modeller and stakeholders to begin with, and the more complex interactions will emerge later as system understanding improves with desk study and more stakeholder interactions.

### *Step 3: Workshops and stakeholder participation*

Nearly all conceptual models that we reviewed integrated some form of stakeholder input, which was variable depending on the decision context and complexity of the method chosen. Causal Loop Diagrams (e.g. Coletta et al., 2024) and Impact Chains (e.g. Sett et al., 2024) for example, generally elicit the integration of more stakeholder input than Influence Diagrams (e.g. Mühlhofer et al., 2023), which have a strong quantitative component. A key step in our approach was to draw on diverse knowledge from a range of expertise, which we did through application in test cases. This way, the Impact Web would be co-created to develop a mutually agreed upon visual output of complex risks, as well as a shared heuristic of the system. Building on the preliminary number of mapped elements in Step 2, we held two workshops in each test case. The workshops were done with range of different stakeholders from communities, policy, practice, civil society, academia and governments. These stakeholders were identified in the scoping (Step 1). Workshop 1 focused on identifying new elements for the Impact Web, as well as reviewing the ones that had already been identified and mapped from the desk review (Step 2). After workshop 1, we included the new elements into the model, and held a second workshop to re-validate the logic and elements, as well as look at entry points for risk management. This stakeholder backstopping provided better understanding of otherwise unknown or missed model elements and their connections, and helped to characterised the complex risk characteristics that could not be captured through desk study alone.

### *Step 4: Review of model and visualisation*

After collecting stakeholder inputs across the five test cases, an important step was to review the model among the research authors. This included in-depth structuring of the information gathered in the

workshops and cross referencing it from available literature sources gathered in the desk study. Where possible, we also refined the number of elements, for example by clustering two elements that represented the same issues. This was done to reduce the model's complexity and ensure the final visual could be an effective communication tool. We also reviewed causal connections and logic behind them, reflecting to understand what this meant in a systems context, thus enhancing our own understanding of complex risks. We then reworked the graphical design to create a visual and causal connections which could be simpler to follow.

### Step 5: Drafting narrative storyline

As a final step to accompany the Impact Web model, a narrative risk storyline was drafted for each test case that described the model and its connections in a narrative format. This helped to communicate in a descriptive and engaging manner the complex model output that results from following the previous steps, making it more engaging and useful to direct risk management decisions for both experts and non-experts (Hanf et al., 2025; van den Hurk et al., 2023). The storylines were drafted by the research authors, explaining the key aspects in the Impact Web and findings from the complex risk assessment. This was done after the authors had completed the desk review, stakeholder workshops and reviewed the model.

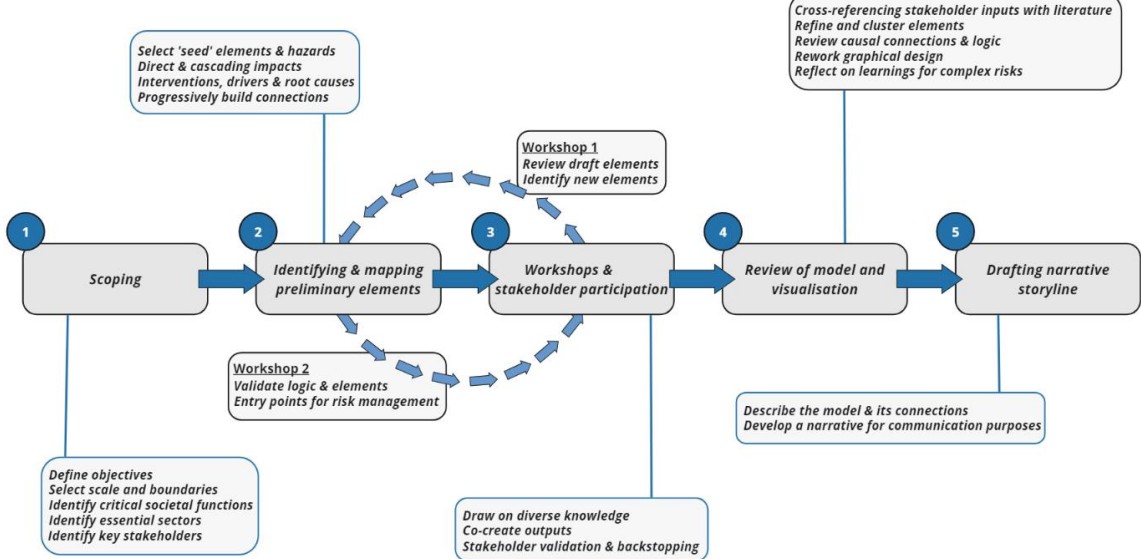

**Figure 2:** *Workflow of the steps that were followed for constructing an Impact Web. We trailed the approach in 5 test cases, which allowed for adjustment and improvement of the methodology as well as stakeholder feedback. The workflow followed a flexible stepwise methodology in five steps (scoping, identifying & mapping a preliminary number of elements, workshops and stakeholder participation, reviewing the model's logic and visualisation and drafting an accompanying narrative storyline). Workshop 1 allowed for new inclusions and adjustment of already identified elements in the draft model. Once included, workshop 2 allowed for validating the logic and looking at entry points for risk management. This is shown in the figure through the circular blue arrows, which indicates iteration in the model's development.*

## 3.    Results: proof of concept

Here we present our result, showing our proof of concept detailing the final output of an Impact Web and narrative storyline from the Guayaquil, Ecuador test case. Below we elaborate on why Guayaquil was chosen for our proof of concept in this paper.

### 3.1.    Complex risks linked to COVID-19, concurring hazards and responses in Guayaquil, Ecuador

Here we show our proof of concept, presenting the results and final outcome of one of the test cases,
from Guayaquil, Ecuador. We only show the results of one case in this paper as our aim has been to
demonstrate how we developed the methodology. Selecting Guayaquil to showcase Impact Webs
highlights the outcomes of steps 4 and 5 in Figure 2.

### Step 1: Scoping

We developed an Impact Web to study risks and impacts emerging from the COVID-19 pandemic and
concurrent hazards, threats and shocks in the city of Guayaquil, Ecuador. Guayaquil was selected due
to its high population density, high levels of poverty and inequality, its large informal work sector,
overcrowded housing (Delgado et al., 2018) and high exposure to climate related and geophysical
hazards (Hallegatte et al., 2013). These factors make the city's inhabitants vulnerable to the
compounding effects of multiple hazards, and presents challenges for risk management that are
exacerbated by limited financial resources at both the municipal and national level. These factors
additionally have numerous and compounding drivers of risks and root causes making it an important
case in which to undertake a complex risk assessment. We used COVID-19 as the 'seed' element for
developing the Impact Web as the hazard has been so diverse in its effects across communities, sectors
and economies, which additionally provided important lessons for the application of a novel conceptual
risk modelling approach using a systems lens. It was decided that taking a case study at the city scale
supported in defining system boundaries and decision context, for which COVID-19 has been cross-
scale and highly dynamic (Hagenlocher et al., 2022).

### Steps 2, 3 & 4: Impact Web of Guayaquil, Ecuador

Figure 3 presents the final conceptual model of the complex risk assessment in Guayaquil. The Impact
Web visualises (i) multiple interacting hazards, threats and shocks across various scales, (ii) the
identification of different risks/impacts for communities, sectors and societal functions as well as their
interconnections and cascading effects (iii) their underlying risk drivers as well as (iv) the root causes
behind underlying risk drivers, some of which can be spatial and temporally distant from newly emerging
risks/impacts. Further, the Impact Web model also maps (v) risks and impacts linked to responses (e.g.
policy interventions aimed to reduce risks) as well as (vi) risks that did not manifest due to the
interventions.

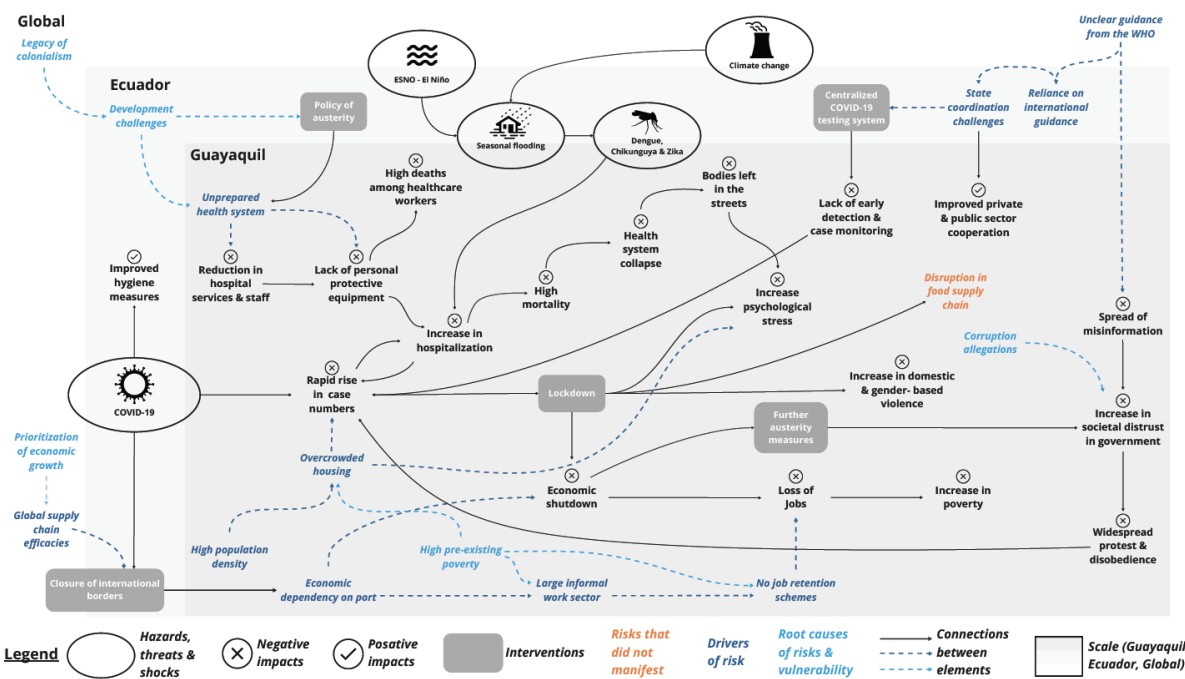


**Figure 3:** *Impact Web for the test case of Guayaquil, Ecuador. The conceptual model visualises*
*complex risks and impacts linked to the COVID-19 pandemic, concurring hazards and the responses*
*to it, as well as interconnections between system elements and drivers and root causes of risks.*

479                    **Step 5: Narrative storyline for Guayaquil, Ecuador**
*The first confirmed case of COVID-19 in Guayaquil was 29 February, 2020. Driven by the city's high*
*population density, challenges with overcrowded housing and unpreparedness in the health system,*
*there was a rapid rise in cases and hospitalizations. Governmental policies of austerity in the five years*
*prior to the pandemic meant that hospitals and healthcare facilities were understaffed and under*
*equipped. The lack of personal protective equipment resulted in a high number of cases and deaths*
*among healthcare workers, which put further pressure on a health system that was already burdened*
*by increases in vector borne diseases due to seasonal flooding exacerbated by climate change. From*
*the compounding effects of multiple hazards and cascading impacts that emerged, the health system*
*reached a tipping point and collapsed, tragically resulting in a large number of bodies being left in the*
*streets, hospitals and care homes. This significantly increased psychological stress for the city's*
*residents. In March of 2020, Guayaquil had an excess mortality rate five times that of the same month*
*in the previous year and the highest COVID-19 mortality rate of any Latin-American city.*

*Economic disruptions from the intervention to close international borders were particularly severe in*
*Guayaquil due to the city's high dependency on the port. The closing of borders triggered economic*
*shutdown, with widespread adverse effects on employment and livelihoods. Due to the lack of job*
*retention schemes, many citizens, a lot of whom were already living in poverty before the pandemic,*
*were left without income generating opportunities. These impacts were exacerbated for the large*
*informal employment sector in Guayaquil. Due to the limited availability of space per person driven by*
*the high population density and overcrowded housing, lockdown interventions and social distancing*
*were difficult to follow for a large segment of the population. As seen in many places, there were also*
*sharp increases in domestic and gender-based violence during lockdown. As Guayaquil is a food-*
*producing country, one risk that did not manifest as a result of lockdowns was disruption in the food*
*supply chain and food shortages that were prevalent in some other countries in the region.*

*State coordination challenges and reliance on international guidance, which was unclear and*
*contradictory in the early stages of the pandemic, meant there was a lack of an integrated, cross-*
*sectoral and multi-scale response between Guayaquil's and Ecuador's public institutions. The national*
*government maintained a centralised COVID-19 testing system, which hindered the effectiveness of*
*city institutions to set up early-detection and monitoring systems such as contact-tracing and testing*
*facilities. The unclear guidance from the World Health Organisation resulted in the output of unclear*
*information and the national level, which was one of the factors that contributed to the spreading of*
*misinformation throughout digital networks. One positive impact that arose from state coordination*
*challenges was the strengthening of public and private sector cooperation.*
*In response to the economic disruptions, the government of Ecuador brought in more austerity*
*measures. Furthermore, corruption allegations were brought against some city and state-level actors*
*for capitalising on the emergency healthcare situation. These factors saw increasing societal distrust in*
*the government, which was already underlying. This came to fruition in Guayaquil when a societal*
*tipping point was reached in May of 2020, resulting in widespread protest and civil disobedience.*
*Key risk drivers identified included initial unpreparedness in the health system, high population density*
*and overcrowded housing, economic dependency on the port and state coordination challenges linked*
*to reliance on international guidance among others. These risk drivers influenced cascading response*
*risks, including widespread negative economic effects of lockdown and closures of international*
*borders, and an increase in societal distrust and subsequent protest and civil disobedience, which were*
*in part due to further austerity interventions in response.*

*A number of considerations for risk management emerged from developing the Impact Web for Guayaquil. These include focusing attention, resources and efforts towards multi-sectoral and multi-scale coordination across public and private institutions, as well as ensuring strong reach and availability of social protection mechanisms and investment in risk monitoring and data systems. The case also highlights that clear guidance and risk communication are key to building societal trust during times of crisis.*

## 4.    Discussion

With Impact Webs, we integrated Climate Impact Chains with aspects of system mapping approaches. Doing this aimed to close gaps in current conceptual models of risks, through characterising dynamic interactions between hazards, exposure, vulnerability, response risks drivers and root causes (IPCC, 2023), improving our understanding of complex risks through following a flexible stepwise methodology. In the discussion we reflect on strengths and limitations of making Impact Webs. While in this paper we only present one test case, our discussion reflects on lessons we learned through the process of developing the methodology and from across all of the cases. We also provide future research directions.

### 4.1.    Strengths

The application of the Impact Web methodology in case studies showed that the approach is useful to conceptualise, identify and visualise networks of interconnected elements across different systems and sectors. The conceptual model's suitability to map the interactions of multiple, concurrent hazards with multiple pre-existing drivers of risk and root cases helps to uncover underlying societal vulnerabilities, and is useful to derive storylines of how interconnected risks and impacts emerge from a hazard or shock events. In the context of Guayaquil, the Impact Web and narrative storyline characterises how COVID-19 revealed vulnerability in the health system, resulting in lockdowns that subsequently affected many other systems and exacerbated already existing economic, domestic, governance challenges in the city and country. Taking COVID-19 as the 'seed' element for our Impact Web resulted in constructing a more simplistic cause-effect chain at the beginning of the modelling exercise, which can be useful for replicability. Given the models effectiveness for mapping an event as complex as COVID-19 suggests that you could equally develop an Impact Web to understand complex climate change risks. Moreover, modelling five test cases with a flexible approach towards the 'local scale' (e.g. a humanitarian camp in Coxes Bazar, a city scale in Guayaquil, a regional focus in Togo and the Indian Sundarbans and a national scale in Indonesia) suggests that you could create an Impact Web to meet needs for a variety of decision contexts. For example, one could create the model to assess complex risks for a river basin, town, or even a specific community.

Applying a systems lens towards analysis and mapping elements in the conceptual model, the developer of an Impact Web as well as the stakeholders engaged gain a more comprehensive overview of complex risks in the system they are mapping. While the final visual and the narrative storyline is the output, it is the process of developing an Impact Web that stimulates critical reflection in the modeler and involved stakeholders, which is one of the key outcomes. In each of the test cases, many of the stakeholders involved in the workshops entered with expertise in one specific sector or to share their own lived experience. However, many participants gave verbal feedback that after collaboratively working on the model together in the workshops, they had learned from other and now better understood impacts and drivers outside of their area of expertise, and thus had a better understanding of complex and cross-sectoral risks. Moreover, involving stakeholders throughout the modelling process can help identify key agents who can act as a catalyst for change (Renn et al., 2022; Özesmi and Özesmi., 2004). These can be, for example, stakeholders who perceive more causal relationships or options to change the system. Working with stakeholders to co-create the model can widen the lens for identifying critical elements, such as feedback effects and trade-offs, which can then be further analysed. Additionally, taking a participatory or bottom up approach for the risk assessment brings in perspectives that can influence top-down decision making.


As the conceptual model not only accounts for negative impacts, but also how policy responses and
societal reactions to policies can lead to additional positive outcomes, as well as unintended
consequences, i.e. risks arising from responses (Simpson et al., 2021), Impact Webs are useful to
reflect on positive and negative outcomes of previous disaster risk management practices. The inclusion
of interventions and response risks and impacts additionally allows for the identification and
management of trade-offs or maladaptation that can occur through decision making processes. While
the outputs of an Impact Web do not quantify the severity or probability of such trade-offs, the approach
is informative by revealing sometimes unclear or more nuanced relationships between decisions and
negative outcomes in the system you are analysing. The visual and accompanying narrative storyline
can thus inform policy and risk management by learning from past impacts, and how these have or
have not disrupted critical societal functions (Hanf et al., 2025). They are additionally effective for pre-
intervention evaluation and for communication purposes (Termeer et al., 2017; Wiebe et al., 2018).
**4.2 Limitations**
Given the complexity of interconnected systems and the ambiguity of system boundaries, it is not
possible to characterise all interconnections using Impact Webs. These models are a simplification of
reality and only the most prominent outcomes are derivable. These prominent outcomes are shaped by
the developers own inherent biases, although the participatory approach aims to reduce this by
providing a mutually agreed upon heuristic of complex risks in a system. In consideration of this, it is
important to acknowledge that participatory modelling is an exercise in which power dynamics come
into play. Therefore, this should be considered when identifying key agents as catalyst of change.
Communicating that the model is a simplification of real-world interactions, as well who it was developed
and with to decision makers is important, to ensure these factors are considered for in policy making.
Even though we recommend standardized constitutive elements and steps for construction, given the
sheer variety of effects originating from one or multiple hazard events, no one Impact Web would be
replicable even if it was developed for the same hazards at the same scale and focus if done by different
stakeholders. Where to define the boundaries of the systems being mapped is vague, and which
elements are selected for the model depending on stakeholders views on key protection targets and
societal functions. A system is usually defined according to its elements within defined system
boundaries (i.e. endogenous system elements) and outside of its boundaries (i.e. exogenous system
elements) (Sillmann et al., 2022) which are selected based on the scale and objectives of analysis.
However, given that we developed a model with COVID-19 as the seed element, which affected all
corners of society and did not occur within defined boundaries, it was difficult to know where to stop.
This challenge could equally arise for developing Impact Web in a multi-hazard multi-risk climate
change context, where the cascading impacts of events are also felt across sectors and scales (van
den Hurk et al., 2023). This 'messiness' of complex and ongoing cascading effects that the Impact Web
sheds light on is a challenge for policy, which often requires sectoral and spatially defined targets, and
equally can render the direct visual output of an Impact Web difficult to engage with.

An additional challenge regards how the outputs of the conceptual model can be integrated with
quantitative data for further analysis. While the logic for our model drew inspiration from reviewing data
driven models including Fuzzy Cognitive Maps, Influence Diagrams and Bayesian Belief Networks, our
approach instead combines stakeholder inputs, desk review and the outcomes of historic events to
arrive at characterisation of how the system of investigation has been affected. As data limitations are
often a challenge when modelling socio-ecological systems, analytics on interactions in a multi-hazard
context would be difficult.

**4.2.    Future research directions**
A number of questions emerge from the application of our methodology that would benefit from further
research. Following the steps for construction enhanced our own understanding of complex risks in the
systems under investigation, and the outputs are useful to communicate complexity. However, a
number of modelling considerations remain to be explored that are important for disaster risk
management, such as temporal dimensions, critical vulnerability moments (de Ruiter & van Loon, 2022)
and system tipping points (Lenton et al., 2023). Bridging conceptual models with quantitative modelling
approaches, as well incorporating lessons from methods that tackle different aspects of complex risks
in more depth such as vulnerability dynamics (e.g. Albulescu and Armaş, 2024) would be useful in this
regard. Additionally, while the model is affective for assessing risks and trade-offs of interventions, a
more structured, decision focused approach and methodology to see how Impact Webs can provide
comprehensive entry points for disaster risk management and climate change adaptation would be
useful. For example, pathways methodologies have been applied to evaluate risk management
decisions in complex systems (Schlumberger, et al., 2024; Haasnoot et al., 2013, Werners et al., 2021).
Thus, integrating conceptual risk modelling with a pathways approach is one avenue that warrants
further exploration. Understanding and mapping risk complexity is only useful if cascading effects and
systemic risks can be minimised, for example through decoupling unnecessary connections across
sectors. Moving from complex risk assessment to complex risk management needs further attention in
order strengthen the resilience of systems.
## 5. Conclusions
This paper offers a new conceptual modelling approach called Impact Webs which identifies,
characterises, and maps complex risks. The inadequacy of single-hazard and single-risk approaches
in the face of global challenges like COVID-19 and climate change emphasizes the need for
comprehensive risk assessment that account for interconnectivity. Impact Webs are one such
methodology in an emerging field of research that do this. Their application in test cases identified
critical links between multiple hazards, responses to them, drivers of risk, root causes as well as pre-
existing societal vulnerabilities. The conceptual model provides a more nuanced understanding of how
risks propagate through systems, offering valuable insights into potential feedback effects, trade-offs,
and key agents that can act as catalysis of change and influence risks in a system. While the approach
contributes to improving complex risk assessment, a number of future research directions presented in
this article would further advance the methodology. These include bridging the conceptual model with
data-driven approaches and transitioning from complex risk assessment to complex risk management
that strengthens systemic resilience. In the evolving and interconnected landscape of communities and
societies, disaster risk reduction and climate change adaptation must account for complexity. The
Impact Webs approach stands as one valuable contribution to realise this, offering a system-wide
perspective for complex risk assessment.
**Data availability.** The data can be provided by the authors upon reasonable request.
**Author contribution.** ES: conceptualization, methodology, formal analysis, writing – original draft
and visualization. DC: conceptualization, methodology, formal analysis, writing – original draft and
visualization. AV: investigation, formal analysis and visualization. SW: conceptualization, methodology,
writing – review & editing, visualization. MH: conceptualization, methodology, formal analysis, writing –
review & editing, visualization.
**Competing interests.** The authors declare that they have no conflict of interest.
**Disclaimer.**
**Special issue statement.**

**Acknowledgements.** This publication has been developed within the CARICO ("Understanding Systemic and Cascading Risks: Learnings From COVID-19") and CARICO SADC ("Lessons from the COVID-19 pandemic for understanding and managing cascading and systemic risks in the SADC region") projects. Views and opinions expressed are those of the authors only and do not necessarily reflect those of UNDRR, GIZ or BMZ.

**Financial support.** The CARICO project has received financial support from the UN Office for Disaster Risk Reduction (UNDRR) and the Government of Germany, notably the Federal Ministry for Economic Cooperation and Development (BMZ) and the Gesellschaft für Internationale Zusammenarbeit (GIZ) GmbH. The CARICO SADC project (Grant Agreement 81292321) is funded by BMZ and supported by GIZ.

**Review statement.**

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
