# Peer review of "Impact Webs: A novel conceptual modelling approach for characterising and assessing"

_EGUsphere, 2024_

## Referee Comment (RC1)

Title: *Impact Webs: A novel conceptual modelling approach for characterising and assessing complex risks*

Authors: Edward Sparkes, Davide Cotti, Angel Valdiviezo Ajila, Saskia E. Werners, Michael Hagenlocher
* * *
**General comments:**

The article presents a novel conceptual complex risk assessment methodology named 'Impact Webs'. The authors describe the conceptualization of this new approach to complex risk modeling and how they developed it based on combining the advantages of several existing conceptual risk modelling approaches. The steps for constructing such a 'Impact Web' are also described, and at the end the authors show the results of a complex risk assessment using a case study in Guayaquil, Ecuador to provide proof of concept.

In general, the structure of the paper is rather unfortunate chosen or the headings are simply inappropriate. The authors describe in Section 2 "Methodological development" that they did a literature review and how this inspired the development and conception of their new 'Impact Webs'. Furthermore, they write that they carried out further theory and conceptual synthesis within the research team (but no further information is provided) to develop the concept for the impact webs. They also briefly mention that they tested the concept with stakeholders in five different case studies. And then in Section 2 "Results" the structure of these 'Impact Webs' and the method for constructing such 'Impact Webs' is described. I think that this Section 3 describes the actual method and not the results. And Section 2 is more of a chapter that describes the methodological pre-considerations. And in Section 3.3, an example is given of how such a development of an 'Impact Web' is carried out using a case study. For me, this is rather a "Results" Section, presenting the results of actually applying the new approach in a case study. Overall, I suggest reconstructing and renaming the sections 2 and 3 including the subsections.

While I find this new 'Impact Webs' approach very exciting and interesting and valuable, I think it has not been sufficiently presented by the authors. Some work needs to be done to make it possible to replicate the method in future studies. Please see my specific comments for more details.

There are also some typos in the manuscript. I would advise the author to go through everything carefully again.

Overall, I recommend major revisions.

**Specific comments:**

1. The authors mention the term 'response risks' in the abstract and later in several places in the manuscript. In lines 52 and 228, they mention this in connection with the literature. But this term is not used in this literature. What do the authors mean by this? Do they mean "human responses", "response options" or "response actions"? Or do they mean here "responses" as a driver of risk (risks due to human responses that not achieve the intended outcome etc.)? If they are introducing a new term, then it should be identified as such.

Otherwise, the use of direct quotation marks in line 228 is confusing, because, as stated, the term as used by the authors here is not mentioned in the cited literature.

2. In lines 32-33 the authors talk about "both positive and negative outcomes of disaster risk management practices". What do you mean by that? I suggest rephrasing it to better clarify what you mean by positive and negative outcomes.

3. In lines 80–82, the authors cite literature on other system mapping approaches on which they base their newly developed Impact Webs: "… such as Causal Loop Diagrams (Coletta et al., 2024), Fuzzy Cognitive Maps (Gómez Martín et al., 2020) and Bayesian Belief Networks (Scrieciu et al., 2021)." It would be good to provide further references per approach and to put "e.g." in front of them, since these are only mentioned as examples of many others.

4. When describing the methodological development of the Impact Webs, the authors mention on line 116 that they "show other conceptual risk modelling approaches" that they drew inspiration from, without mentioning, however, that this overview was based on a literature review. It would be helpful to make this clear at this point, since the authors also name the next subsection accordingly.

5. In lines 121 to 127, the authors explain their literature search conducted as part of the study. But the information given about the process is much too thin. The method is not sufficiently documented to allow a replication of the review. Which search string was used? In which database was the search conducted? Why was this database selected and not others? Was the search restricted to peer-reviewed literature? How many articles were screened in total? What are the exclusion and inclusion criteria? Why was no systematic literature search conducted?

6. In Table 1, the authors provide an overview of conceptual risk modelling approaches. I find this table very helpful. I am just wondering what the different approaches are categorized by? The authors start with "Climate Impact Chains" and end with "Participatory System Mapping". It might make sense to categorize the approaches by the degree of integration of quantitative data or the approaches' ability to capture the complexity of the system (from linear to non-linear approaches). Furthermore, I wonder what the key references in the last column are sorted by? Relevance? Does it matter who is listed first? I would suggest sorting them by year of publication.

7. Furthermore, I doubt that the last category presented by the authors "Participatory System Mapping" is not really a stand-alone conceptual model-risk model development approach. Participatory systems modelling can be done using various systems mapping methods (e.g. FCM or causal loop diagrams) to formalize knowledge. The mapping of conceptual systems can be done both participatory or on the basis of literature research. "Participatory" merely describes the process and the way in which the knowledge is generated. All of the first five approaches could be carried out in a participatory way. See, for example, the studies:
   - Videira et al. 2009. Scoping river basin management issues with participatory modelling: The Baixo Guadiana experience.
     https://doi.org/10.1016/j.ecolecon.2008.11.008
   - Sahin et al. 2020. Developing a Preliminary Causal Loop Diagram for Understanding the Wicked Complexity of the COVID-19 Pandemic.
     https://doi.org/10.3390/systems8020020

- Melles et al. (2021). COVID 19: Causal Loop Diagramming (CLD) of Social-Ecological Interactions for Teaching Sustainable Development. In: Leal Filho, W. (eds) COVID-19: Paving the Way for a More Sustainable World. World Sustainability Series. Springer, Cham. https://doi.org/10.1007/978-3-030-69284-1_16
- Olazabal et al. 2018. Transparency and Reproducibility in Participatory Systems Modelling: the Case of Fuzzy Cognitive Mapping. https://doi.org/10.1002/sres.2519

8. The text in lines 131-173 in subsection 2.1 on the "Lessons from the review" does not really fit with the results presented in Table 1. The authors introduce three broad types of approaches following a study by Elsawah et al. (2017) in this section. However, the connection to the categories the authors have identified from their review (presented in Table1) is not entirely clear. I do not see the benefit of introducing this broad classification by Elsawah et al. (2017). The examples in the table are not organized according to this classification. It just seems so disconnected. It would be better if the authors simply focus on the lessons learned from their own review and discuss and compare the strengths and weaknesses of the identified methods, in order to derive what they used for their new approach. The authors do this from line 161 onwards, but it would be interesting if the authors elaborate on this further. I would advise the authors to shorten the section (L132-173) and focus more on the actual results of their review.

9. In line 172 to 173, I do not understand what the authors mean with "of what stakeholder value and want to protect." I am not sure what the word "protect" means in this context. Please consider revising the sentence.

10. In section 2.3, the authors describe the next steps in how they made this methodology feasible for practical application. They briefly state that they tested the methodology with groups of stakeholders in various case studies. What I am missing here is a better overview of the five different case studies. Simply saying that you have five different case studies and giving their names is not enough. Why were five case studies selected? Why these in particular? What characterizes each case study? A table would be nice that provides an overview of the uniqueness and challenges (including the complex risks) of each case study in comparison to the others.
What did the authors learn from each case study?
The authors further describe that there were two workshop rounds with a number of different stakeholders. Were there two workshop rounds in total or two workshop rounds for each case study? This is not entirely clear.

11. In line 179, the authors mention that the identified various theories and concepts were synthesized within a "research team". However, the authors do not provide any further details about this research team. It would be useful to get a clearer sense of how this team and this synthesis process was organized. How big was the team? Did all belong to the same institution? What was the team composition? What disciplines were represented in the team? What about the disciplinary bias of the team? Were there members of this team who led this conceptual development? What was the expertise of these members? And how long did the process of this conceptual development take?

12. As already described in my general comments, I suggest renaming Section 2 and restructuring and renaming Section 3. The current structure is misleading. Thew way I understand the text, Section 2 contains only preliminary methical considerations (for

example, a preparatory literature search), while Section 3.1-3.2 describe the actual new method/approach of 'Impact Webs'. In fact, I see section 3.3 as a "Results" Section in which the application of the new approach is presented and tested in a case study.

13. The authors present in Section 3.1 the constitutive elements of an 'Impact Web'. Unfortunately, this is not very illustrative. It would be much better if the authors present these eight model elements in a separate table directly next to Figure 1. It would also be important for the recognition of the individual elements that the elements are not grouped together or named differently. For example, the authors have combined "Interventions & response risks" and "Risks that did non manifest". I would strongly advise against this. It would also be good to use the same colors for the elements in such a table as in Figure 1.

14. For the model element "Connections between elements", please explain the syntax in more detail. It is not entirely clear from the text what the different line types (solid, dashed) mean.

15. For the model element "Multiple interacting hazards", please indicate that you use icons in the model.

16. For the model element "Direct & cascading impacts", please explain that you distinguish between positive and negative impacts, which are indicated by cross and hook signs. Why did the authors choose these symbols? Wouldn't plus and minus signs be more intuitive?

17. The authors mention in L292 that during the scoping step an important step for selecting the scale to model by looking at geographical or administrative boundaries. It would be interesting to learn more about when the authors would suggest considering geographic or administrative boundaries. Risks can cross administrative boundaries. What would be the argument for looking at administrative boundaries anyway?

18. L306-307: I don't get what you want to say with this sentence: "This perspective acknowledged that the systems relationships emerge more clearly when under stress, i.e. become more visible and therefore easier to observe." What do you mean with "under stress"? Please consider revising this sentence.

19. In lines 364 to 367, the authors mention that they only select one case study to demonstrate the use of the new 'Impact Webs' approach. I fully understand why only one case study was selected. But why did the authors choose the Guayaquil case study and not one of the others? It would be good to provide a justification.  Now, reading on, I see that the authors state in Section 3.3. "Step 1: Scoping" why they chose this case study. But perhaps it would be good to make this point earlier.

20. In lines 386 to 394, the authors summarize the results from step 2,3 and 4 for the Guayaquil case study. But actually, they only mention which elements are included in the final model. That provides no added value. The reader learns nothing about the process, how the elements were selected, how the workshops with the stakeholders went, whether there was disagreement, or how many elements there were initially and how many the model was reduced to in the end? More details about the actual steps 2, 3 and 4 would be extremely helpful for further future applications of the 'Impact Webs'. And also, how long did each step take? Please provide further details in this regard.

21. In "Step 5: Narrative storyline for Guayaquil, Ecuador", the authors provide the storyline they have developed for this case study. The story is very interesting and important for the later communication with stakeholders. But wouldn't it also be interesting for the reader to learn more about the process of writing this story? Who of the team wrote it? How was the story constructed? Was an overarching storyline followed? Were there difficulties? How often was the story reflected upon with the entire team? How long did the process take?
It would also be good to highlight the "actual story" in the text, for example by using italicized font. If I understand it correctly, the actual story goes from line 402 to 438. Or am I wrong?

22. Figure 3: It would be important if the elements were listed in the same order in the legend as in Figure 1, which presented the basic structure of an 'Impact Web'. It would also be important to use the same terminology. In Figure 3, for example, the authors only refer to "Root cause" instead of "Root cause of risk and vulnerability".

23. In lines 439 to 451, the authors highlighted and described the advantages of the 'Impact Web' approach in the case study of Guayaquil. But that doesn't fit into "Step 5: Narrative storyline for Guayaquil, Ecuador" at all, does it? Wouldn't it be better in a separate paragraph? Somehow it is a bit misleadingly placed.

24. In Section "4.1 Strengths", the authors say that the new approach "is useful to conceptualise, identify and visualise networks of interconnected elements across different systems and sectors." To prove this, it would be good to get a brief overview of the challenges and successes and lessons learned from the other case studies. A short overview table would be good for this. It might also be helpful to show the other impact web models of the other 4 case studies in the supplementary material.

25. L472-473: "Given the models effectiveness for mapping the complexity of an event such as COVID-19 suggests that you could equally develop an Impact Web to understand the complexity of climate change risks." I don't understand this sentence. Is it related to what was said in the previous sentence, that a simple cause-effect chain model was developed first?

26. In lines 482 to 484, the authors state that, apart from the outputs (the visual and the narrative storyline), it was rather the process of developing the model that stimulated critical reflection in the modeler and involved stakeholder. Unfortunately, however, we did not learn much about this process when the example of Ecuador was presented. Please elaborate on this in an appropriate place.

**Technical corrections:**

There are some grammatical errors and typos. I have highlighted the ones I noticed, but I suggest that the authors read the manuscript carefully again.

1. L29: "the methodologies usefulness" → "the methodologies' usefulness"
2. L87: "compared with Climate Impact Chains" → "compared **to** Climate Impact Chains"
3. L116: "In section 2, we present our methodological development from Impact Webs." → "In section 2, we present our methodological development **of** Impact Webs."
4. L118: Is the word "trailed" really appropriate in this context?

5. L179: There is a typo: "until we synthesized an agreed" → "until we synthesized an**d** agreed"
6. L217: "Following the inclusion multiple hazards" → "Following the inclusion **of** multiple hazards"
7. L267: There is a typo: "For exmaple" → "For e**xa**mple"

---

## Author Comment (AC4)

**Reviewer 2**

In the manuscript "Impact Webs: A novel conceptual modelling approach for characterising and assessing complex risks", the authors make use of expert judgment and non-systematic literature review to build on lessons from different approaches to come up with a new conceptual modelling approach for systemic risk characterization. It is a relevant topic, however the contextualization and purpose of the paper are not clearly supported. I would thus recommend major revisions as outlined in the following.

*Authors response: Thank you for your useful comments on our draft publication. We are glad you think it is a relevant topic. We feel that, upon integrating reviewer feedbacks, it will add value to the peer reviewed literature on complex risk assessments.*

**The structure** of the manuscript is very unclear. It is unclear what is result, what is method, and most often, how certain conclusions were drawn. The authors should revisit their structure and narrative for this paper. It is not clear to me, what this paper offers to its readers. Is it the pure idea of impact webs (as an advancement of impact chains), is it the visualization, is it the guidance to build impact chains? In its current form, the authors seem to do everything a bit, but nothing sufficiently in depth. Re-structuring and clarifying the objective of this paper. Just outlining the process of developing the web (as mentioned in the introduction), seems to fail answering a specific research question.

*Authors response: Our aim with this publication is to offer a new methodology to improve understanding of complex risks. We agree that this could be made clearer in the introduction, and will do so in a second submission by setting out the aim clearly. To demonstrate how we achieved this aim, in submission 1 we offered the methodology of Impact Webs and guidance in how we made it as 'the result' – however based on you're and the other reviewers feedback we will significantly restructure sections 2 and 3 of the paper as follows: 1) Introduction (with more focus on the aim), 2) Methodology, 2.1) Methodological pre-development: Scoping review of conceptual risk models for inspiration, 2.2) Selection of elements, 2.3) Steps for constructing an Impact Web, 2.4) Trail in test cases (with more details of the cases), 3) Results: Proof of concept, 3.1) Complex risks linked to COVID-19, concurring hazards and responses in Guayaquil, Ecuador. We will then keep the discussion in the same structure, but reflect more on the aim and research gap that we closed with Impact Webs. Please see the restructured outline below:*

1. *Introduction*

2. *Methodological development*

    *2.1 Methodological pre-development: Scoping review of conceptual models of risk for inspiration*

    *2.2 Selection of constitutive elements in an Impact Web*

    *2.3 Steps for constructing an Impact Web*

    *2.4 Trail in test cases*

3. *Results: Proof of concept*

    *3.1 Complex risks linked to COVID-19, concurring hazards and responses in Guayaquil, Ecuador*

4. *Discussion*

    *4.1 Strengths*

    *4.2 Limitations*

    *4.3 Future research direction*

5. *Conclusions*

*With this new structure, and additional details on the methodological development, the manuscript will be clearer offer to its readers insights on how we developed a new methodology in Impact Webs.*

**What is Impact Webs?** It would be very valuable if authors could clarify what they mean when they refer to impact webs as a conceptual modelling approach. Part of what the authors present seems to be tools (how to visualize), some analysis guidance (see Figure 2). Overarchingly, it would be beneficial, if the authors could make it more explicit, what the purpose of impact webs is. They refer to Bayesian Belief Networks and other modelling methodlogies, include participatory elements which are then refined/complemented through desk studies. Are impact webs meant to be complete and/or correct? Used by who?

*Authors response: With Impact Webs we developed a conceptual model that aim to improve understanding of complex risks in the system or location being modelled. The methodology is flexible and can be applied in the chosen system the modeler wants to investigate. We drew on inspiration from Climate Impact Chains, Bayesian Belief Networks and other conceptual models which are used in risk assessments, which we highlight in table 2 through a non-systematic scoping review. We will significantly restructure the section 'lessons from the scoping review' to show more clearly what aspects of other models inspired us and how we selected different aspects from them for Impact Webs. In the paper we also show how we made impact webs, as its our intention for this to be a methodological paper where readers can replicate our method for their own setting. We can make more explicit in the introduction what we mean when we refer to impact webs, and will lay out more clearly the purpose of Impact Webs in the introduction.*

**Method section:** This paper seems to heavily rely on expert judgment - which makes it very difficult to reproduce and to offer evidence regarding the claims offered here. One key question I had when reading this section was why the authors limited their search for inspiration to the field of single-hazard risk assessment instead of learning from fields that address similar or different complex systems (e.g. integrated water management, agent based modelling, system dynamics research community). If I understand correctly, the authors propose this method based on iterations/refinements in 5 case studies. At least a short introduction of these cases would already offer insight regarding the complex risk context/dynamics Impact Webs has been developed upon I would recommend taking inspiration from studies that have developed methodological approaches or investigated how such approaches have been developed (e.g. McMeekin, 2020) to extend the method section and add an additional section covering the approach development process.

*Authors response: Based on your and the other reviewers' feedback we will significantly restructure the methods section (see above response to the structure). In our scoping review, we did not limit our search to single-hazard approaches. We reviewed and drew inspiration from various other conceptual modelling approaches that are used in risk assessments. Some of these approaches are applied more for single-hazard risk assessments, and some more in multi-hazard or multi-risk assessments. Agent-based modelling was not one of the approaches that we included in the table as we aimed to develop a model that drew on graphical aspects and did not rely on heavy use of computational and quantitative data. Nearly all of the papers we include in table 1 integrate system dynamics and take a systems perspective, which is stated in chapter 2.1, and the reason we did not explicitly draw on integrated water management is because it is not a conceptual modelling approach. We will remove the participatory system mapping row from the table based on the other reviewers' feedback, are can replace this with agent-based modelling if you think this is an important category to include in such an overview of conceptual models of risk.*

*We will also include more details on the 5 case studies in the new section 2.4, and we feel it is best to remove the current section 2.2. 'concept development' as it will make the methodology clearer based on the new structure. While developing Impact Webs, we held a number of internal conceptual development discussions within our team (the Vulnerability Assessment, Risk Management and Adaptive Planning section at UNU-EHS) where we brainstormed and critiqued ideas from one another. While this was done to inspire concept development, the team and synthesis process was not systematically organized. Under the new structure of the paper, where we intend to present the Selection of elements (2.2), Steps for construction (2.3) and Trail in test cases (2.4) in the methodology, we think the approach development process will be sufficiently covered.*

**Regarding the Impact web development process:** Table 1 looks like something that would be worth for the Appendix or could be used in a shortened version to support a discussion of the different methods in the context of complex risk elements to be addressed with Impact Webs (section 3.1?). It would also be interesting to learn, why authors refer to storyline approaches as one of the steps in the impact web development process but did not consider them in Table 1 for inspiration. Section 3.1 seems a mix of presenting the complex risk elements of interest and mentioning what elements from which approach were used to visualize. I would suggest to separate these two purposes and rather provide more justification regarding the choices regarding the visual elements, e.g. by referring visualization research that justifies the choices. I also want to point out that terminology in Figure 1 is inconsistent (and not referred to in the paper). 'driver of risk', 'hazard', 'vulnerability' are all concepts that overlap (at least partially) and thus do not offer clear guidance what visualization element should be used.

*Authors response: We will significantly restructure section 2.1. based on your and other reviewers' feedback, and will highlight the different strengths in the approaches presented in table 1 and discuss why we drew on these for Impact Webs. This would then better support a discussion of the different aspects of approaches that are useful in a complex risk context that inspired us. We did not include storyline approaches in table 1 as they are not conceptual modelling approach. However, we did include storylines as we did not only limit our methodological development to the scoping review of conceptual models, although we understand this may be confusing in how the paper is currently structured in submission 1. In the restructured methods section we feel it will be clearer, and can in elaborate that the scoping review of conceptual models was not our only source of inspiration. We will provide more justification regarding the choices of elements and the visual elements, and will produce a new table next to Figure 1 which includes the elements, how we chose to visualize them, and a short description and key references so there is clarity in terminology. We additionally agree to not group the elements, and will separate them.*

The development steps (section 3.2) is unclear whether they are an outcome of the paper or the method to derive it. It is presented as a method (with limited justification why it is done that way), but lack insights/guidance into how the complexity of systemic risk (and the corresponding visualization) can be managed.

*Authors response: We will restructure the paper so this is now in the methodology – it was our original intention with the paper to present the method (i.e. how to make impact webs) as the result, but will change this based on both reviewers feedback. We reflect and give strengths and weaknesses into how the complexity of systemic risk can be understood and managed through the corresponding visualization and development of impact webs in the discussion section. We feel this is the best situated place for such a reflection in the paper.*

McMeekin, N., Wu, O., Germeni, E., and Briggs, A. (2020). How methodological frameworks are being developed: evidence from a scoping review. BMC Med. Res. Methodol. 20, 173. https://doi.org/10.1186/s12874-020-01061-4

*Authors response: Thank you for providing this reference. This will be useful inspiration for restructured manuscript.*

---

## Referee Report (RR1)

Title: *Impact Webs: A novel conceptual modelling approach for characterising and assessing complex risks*

Authors: Edward Sparkes, Davide Cotti, Angel Valdiviezo Ajila, Saskia E. Werners, Michael Hagenlocher
* * *
**Reviewer 1 | Comments  2nd round**

*Recommendation: I recommend minor revisions.*

I thank the authors for revising their manuscript carefully following my major revisions recommended. The manuscript is almost ready for publishing. I have only some minor revision requests that concern my past comments.

For clarification: From now on (see below), all black texts are my comments from the first round, the red text are the authors' comments after the first round and I have now drawn my new comments for the second revision in blue. When referring to specific line numbers, I am referring to the tracked changes file the authors have resubmitted.
* * *
**Responses to reviewer 1**
**General comments:**

The article presents a novel conceptual complex risk assessment methodology named 'Impact Webs'. The authors describe the conceptualization of this new approach to complex risk modeling and how they developed it based on combining the advantages of several existing conceptual risk modelling approaches. The steps for constructing such a 'Impact Web' are also described, and at the end the authors show the results of a complex risk assessment using a case study in Guayaquil, Ecuador to provide proof of concept.
In general, the structure of the paper is rather unfortunate chosen or the headings are simply inappropriate. The authors describe in Section 2 "Methodological development" that they did a literature review and how this inspired the development and conception of their new 'Impact Webs'. Furthermore, they write that they carried out further theory and conceptual synthesis within the research team (but no further information is provided) to develop the concept for the impact webs. They also briefly mention that they tested the concept with stakeholders in five different case studies. And then in Section 2 "Results" the structure of these 'Impact Webs' and the method for constructing such 'Impact Webs' is described. I think that this Section 3 describes the actual method and not the results. And Section 2 is more of a chapter that describes the methodological pre-considerations. And in Section 3.3, an example is given of how such a development of an 'Impact Web' is carried out using a case study. For me, this is rather a "Results" Section, presenting the results of actually applying the new approach in a case study. Overall, I suggest reconstructing and renaming the sections 2 and 3 including the subsections.
While I find this new 'Impact Webs' approach very exciting and interesting and valuable, I think it has not been sufficiently presented by the authors. Some work needs to be done to make it possible to replicate the method in future studies. Please see my specific comments for more details.
There are also some typos in the manuscript. I would advise the author to go through everything carefully again.
Overall, I recommend major revisions.

*Authors response: We would like to express our thanks to the reviewer for their time to provide useful, constructive and detailed comments. We are glad to hear that they find the Impact Webs approach exciting, interesting and valuable. We agree with the feedback on the structure and have revised the overall structure of*

*sections 2 and 3 of the paper as per your recommendations. Section 2.1 has become the 'Methodological pre-development: Scoping review of conceptual models of risk for inspiration', and we have significantly restructure the 'Lessons from the review' section based on your comments below. The 'Selection of constitutive elements' & 'Steps for constructing an Impact Web' have been made part of the methodology in sections 2.2 and 2.3 respectively, and we have provided more information and detail of the case studies in the 'Trail in test cases' section, moving it to the end of the methodology in 2.4. We have removed the section on concept development (see specific response below). Section 3.3, where we show our proof of concept, giving an example of how such a development of an 'Impact Web' was carried out using a case study is the new results section as per your recommendation. Please see the restructured outline below:*

***1. Introduction***
***2. Methodological development***
***2.1** Methodological pre-development: Scoping review of conceptual models of risk for inspiration*
***2.2** Selection of constitutive elements in an Impact Web*
***2.3** Steps for constructing an Impact Web*
***2.4** Trail in test cases*
***3. Results: Proof of concept***
***3.1** Complex risks linked to COVID-19, concurring hazards and responses in Guayaquil, Ecuador*
***4. Discussion***
***4.1** Strengths*
***4.2** Limitations*
***4.3** Future research direction*
***5. Conclusions***

*With this new structure, and additional details on the methodological development, the manuscript is clearer for replicating the Impact Webs method in future studies. We have responded to your specific comments below.*
***When referring to edits made in response to your comments, we are referring to the tracked changes file we have resubmitted.***

Thank you for addressing my concerns about the structure of the paper. I really appreciate you revising the sections. It's much clearer now.

**Specific comments:**

1. The authors mention the term 'response risks' in the abstract and later in several places in the manuscript. In lines 52 and 228, they mention this in connection with the literature. But this term is not used in this literature. What do the authors mean by this? Do they mean "human responses", "response options" or "response actions"? Or do they mean here "responses" as a driver of risk (risks due to human responses that not achieve the intended outcome etc.)? If they are introducing a new term, then it should be identified as such.
Otherwise, the use of direct quotation marks in line 228 is confusing, because, as stated, the term as used by the authors here is not mentioned in the cited literature.

***Authors response:*** *We use response risks to refer to the risks arising from responses to risks and impacts (e.g responses not achieving their intended objectives, or having trade-offs). The concept and term were introduced in the paper by Simpson et al. (2021), then adopted and extensively used in the IPCC AR6 WGII report, Chapter 1 'Point of departure and key concepts' (e.g. Figure 1.5-part C). In that report, an explanation of response risk is provided in multiple instances, e.g. "The risks of climate change responses include the possibility of responses not achieving their intended objectives or having trade-offs or adverse side effects for other societal objectives", or "risks can arise from potential impacts of climate change as well as human responses to climate change." To address the comment, we have elaborated on the term to explain how it is defined in the literature in the introduction, following the IPCC point of departure and Simpson et al (2021) referencing (lines 53 – 54 in track change file), removed 'risks' from the abstract (line 15) and include reference to AR6 Chapter 1 (line 54). We have additionally removed the quotation marks on line 228 (now line 291) to avoid confusion.*

Thank you for clearing up the confusion.

2. In lines 32-33 the authors talk about "both positive and negative outcomes of disaster risk management practices". What do you mean by that? I suggest rephrasing it to better clarify what you mean by positive and negative outcomes.

*Authors response: This is referring to the methodology being useful to evaluate trade-offs in decision making. We elaborated on this point in the discussion, (lines 652 – 667). To make sure this is clear in the abstract, have update the line to read "The participatory process of developing Impact Webs promotes stakeholder engagement, uncovers critical elements at risk and helps to evaluate trade-offs in decision making by improving understanding of both positive and negative outcomes of disaster risk management practices." (lines 32 – 33)*

I am still not satisfied with the answer. I was concerned about the wording "positive and negative outcomes of disaster risk management practices". What do you mean by "positive" and "negative"? Co-benefits and trade-offs? Please rephrase the wording "positive" and "negative". What is positive and what is negative depends on one's point of view. A more elegant formulation would be desirable.

3. In lines 80–82, the authors cite literature on other system mapping approaches on which they base their newly developed Impact Webs: "… such as Causal Loop Diagrams (Coletta et al., 2024), Fuzzy Cognitive Maps (Gómez Martín et al., 2020) and Bayesian Belief Networks (Scrieciu et al., 2021)." It would be good to provide further references per approach and to put "e.g." in front of them, since these are only mentioned as examples of many others.

*Authors response: Thank you for the suggestion, we have done this.*

Okay. Thank you for the revision.

4. When describing the methodological development of the Impact Webs, the authors mention on line 116 that they "show other conceptual risk modelling approaches" that they drew inspiration from, without mentioning, however, that this overview was based on a literature review. It would be helpful to make this clear at this point, since the authors also name the next subsection accordingly.

*Authors response: Thank you for the suggestion, we have done this.*

Thank you for the revision.

5. In lines 121 to 127, the authors explain their literature search conducted as part of the study. But the information given about the process is much too thin. The method is not sufficiently documented to allow a replication of the review. Which search string was used? In which database was the search conducted? Why was this database selected and not others? Was the search restricted to peer-reviewed literature? How many articles were screened in total? What are the exclusion and inclusion criteria? Why was no systematic literature search conducted?

*Authors response: While systematic literature reviews have many strengths and allow for replication, they are not pre-requisite for conceptual inspiration and for the creation of new methodological approaches. Adopting a non-systematic scoping review approach gave us exploratory flexibility, for example to include and discuss new publications as we came across them, as well as include aspects of methods that we had used and were familiar with in our past research experience. While scoping reviews are not replicable, they are very helpful for developing initial conceptual frameworks and for concept development general.*
*To avoid confusion around this stipulation, we have renamed the section "Methodological pre-development: Scoping review of conceptual risk models for inspiration" to emphasize that our review was a scoping review to support evidence synthesis and inspire concept development. We now also elaborated on why a scoping non-systematic review was chosen in the text below (Lines 137 – 149). We feel that the table presents a useful synthesis of conceptual models used in risk assessments, and has merit in being published as it can be useful for other researchers using these methodological approaches in their work.*

Thank you for clearing this up.

6. In Table 1, the authors provide an overview of conceptual risk modelling approaches. I find this table very helpful. I am just wondering what the different approaches are categorized by? The authors start with "Climate Impact Chains" and end with "Participatory System Mapping". It might make sense to categorize the approaches by the degree of integration of quantitative data or the approaches' ability to capture the complexity of the system (from linear to non-linear approaches).

Furthermore, I wonder what the key references in the last column are sorted by? Relevance? Does it matter who is listed first? I would suggest sorting them by year of publication.

*Authors response: We are glad that Table 1 is found helpful. The different approaches are categorized by how the authors in the selected key references named them. If an author stated in their paper the method they used was Climate Impact Chains or Fuzzy Cognitive Maps for example, we follow their terminology.*
*While we appreciate there can be other ways to categorize the approaches, we do not think it makes sense to overly complicate this categorization in Table 1, for example, by degree of integration of quantitative data or the approaches ability to capture complexity of the system. We highlight in the table where approaches have strengths for integrating quantitative data, and the columns 'Strengths' and 'Weaknesses' in a complex risk context provide information on the approaches ability to capture complexity of the system. While we do highlight methodological ambiguity and cross-over in approaches (lines 200-211), we feel that such a table in a simpler format, categorized by approach name, has value. We propose not to categorize by degree of integration of quantitative data, but will have organized the references by year of publication.*

Okay, I see your argument for not categorizing the approaches. But you haven't addressed the last part of my point 6: "[...] I wonder what the key references in the last column are sorted by? Relevance? Does it matter who is listed first? I would suggest sorting them by year of publication." I am only referring to the **"Key references"** in column 5 of your Table 1. I would sort them in descending order, starting with the most recent publication. For example, for the approach 'Climate Impact Chains':
Sett et al (2024)
Petutschni et al (2023)
Zebisch et al (2023)
Harris et al (2022)
Hagenlocher et al (2018)

7. Furthermore, I doubt that the last category presented by the authors "Participatory System Mapping" is not really a stand-alone conceptual model-risk model development approach. Participatory systems modelling can be done using various systems mapping methods (e.g. FCM or causal loop diagrams) to formalize knowledge. The mapping of conceptual systems can be done both participatory or on the basis of literature research. "Participatory" merely describes the process and the way in which the knowledge is generated. All of the first five approaches could be carried out in a participatory way. See, for example, the studies:

• Videira et al. 2009. Scoping river basin management issues with participatory modelling: The Baixo Guadiana experience. https://doi.org/10.1016/j.ecolecon.2008.11.008

• Sahin et al. 2020. Developing a Preliminary Causal Loop Diagram for Understanding the Wicked Complexity of the COVID-19 Pandemic. https://doi.org/10.3390/systems8020020

• Melles et al. (2021). COVID 19: Causal Loop Diagramming (CLD) of Social-Ecological Interactions for Teaching Sustainable Development. In: Leal Filho, W. (eds) COVID-19: Paving the Way for a More Sustainable World. World Sustainability Series. Springer, Cham. https://doi.org/10.1007/978-3-030-69284-1_16

• Olazabal et al. 2018. Transparency and Reproducibility in Participatory Systems Modelling: the Case of Fuzzy Cognitive Mapping. https://doi.org/10.1002/sres.2519

*Authors response: Thank you for this observation. We agree and have removed the participatory system mapping category from the table. We have incorporated some of the strengths and weaknesses of participatory system mapping approaches in other categories where their methodological application was relevant, as we feel they are important aspects to draw attention to in conceptual risk modelling approaches, and are equally relevant for these other types of approaches.*

Thank you for taking into account my remark and for implementing the advice.

8. The text in lines 131-173 in subsection 2.1 on the "Lessons from the review" does not really fit with the results presented in Table 1. The authors introduce three broad types of approaches following a study by Elsawah et al. (2017) in this section. However, the connection to the categories

the authors have identified from their review (presented in Table1) is not entirely clear. I do not see the benefit of introducing this broad classification by Elsawah et al. (2017). The examples in the table are not organized according to this classification. It just seems so disconnected. It would be better if the authors simply focus on the lessons learned from their own review and discuss and compare the strengths and weaknesses of the identified methods, in order to derive what they used for their new approach. The authors do this from line 161 onwards, but it would be interesting if the authors elaborate on this further. I would advise the authors to shorten the section (L132-173) and focus more on the actual results of their review.

*Authors response: We agree, and have completely restructured this section based on your and other reviewers' feedback. Following your suggestion, we have highlighted the different strengths in the approaches presented in table 1 and discuss why we drew on these for Impact Webs. This has made the section more informative of which aspects of different methods we were inspired by, and more logically link the section to the information in Table 1.*

Okay. Thank you for taking into account my advice and revising the entire sub-section.

I would like to draw your attention to the new study by Hanf et al. 2025 on using causal loop diagramming to derive an integrated, holistic understanding of the complex urban flood risk & climate change adaptation nexus in the coupled socio-ecological system of the city of Hamburg. In addition to the system elements, they have also integrated interventions into their conceptual risk model in order to understand vicious cycles of systemic feedback loops that reinforce flood risk and hinder climate change adaptation processes. It also important to mention that they use a narrative-based approach to communicate the results. By linking the individual feedback stories, they derived an overall system narrative to make the identified feedback structure more concrete and to offer an invitation to an integrated discussion and debate. Perhaps you consider this study worthy of mention.

*Hanf et al. (2025). Towards a socio-ecological system understanding of urban flood risk and barriers to climate change adaptation using causal loop diagrams. International Journal of Urban Sustainable Development, 17(1), pp. 69–102. doi: 10.1080/19463138.2025.2474399.*

9. In line 172 to 173, I do not understand what the authors mean with "of what stakeholder value and want to protect." I am not sure what the word "protect" means in this context. Please consider revising the sentence.

*Authors response: We have revised these sentences to be clearer and provide more of an elaboration of what we mean here (lines 237-242).*

Thank you for taking into account my remark and for implementing the advice.

10. In section 2.3, the authors describe the next steps in how they made this methodology feasible for practical application. They briefly state that they tested the methodology with groups of stakeholders in various case studies. What I am missing here is a better overview of the five different case studies. Simply saying that you have five different case studies and giving their names is not enough. Why were five case studies selected? Why these in particular? What characterizes each case study? A table would be nice that provides an overview of the uniqueness and challenges (including the complex risks) of each case study in comparison to the others.

What did the authors learn from each case study?
The authors further describe that there were two workshop rounds with a number of different stakeholders. Were there two workshop rounds in total or two workshop rounds for each case study? This is not entirely clear.

*Authors response: We have updated this section to give more details of the five case studies, including why they were selected, their characteristics and unique challenges. We have also provided more details on the*

*workshops. As we have restructured the paper, the workshop details are provided in section 2.3. We feel this is the most appropriate place to elaborate on them, given it is where we detail the steps we followed to make Impact Webs*

Thank you for addressing this and for providing a more comprehensive overview on the case studies.

11. In line 179, the authors mention that the identified various theories and concepts were synthesized within a "research team". However, the authors do not provide any further details about this research team. It would be useful to get a clearer sense of how this team and this synthesis process was organized. How big was the team? Did all belong to the same institution? What was the team composition? What disciplines were represented in the team? What about the disciplinary bias of the team? Were there members of this team who led this conceptual development? What was the expertise of these members? And how long did the process of this conceptual development take?

***Authors response:*** *We feel it is best to remove this section as it will make the methodology clearer based on the new structure suggested by the reviewers. While developing Impact Webs, we held a number of internal conceptual development discussions within our team (the Vulnerability Assessment, Risk Management and Adaptive Planning section at UNU-EHS) where we brainstormed and critiqued ideas from one another. While this was done to inspire concept development and is part of any research work, the team and synthesis process was not systematically organized. Under the new structure of the paper, where we present the selection of elements and steps for construction in the methodology, rather than as results, we do not feel this section adds much to the reader.*

Okay.

But in L96 you still speak of a "research team". Either rephrase it without using 'research team' or I suggest adding a sentence that explains who the team is and how it is composed. Thank you.

12. As already described in my general comments, I suggest renaming Section 2 and restructuring and renaming Section 3. The current structure is misleading. The way I understand the text, Section 2 contains only preliminary methodical considerations (for example, a preparatory literature search), while Section 3.1-3.2 describe the actual new method/approach of 'Impact Webs'. In fact, I see section 3.3 as a "Results" Section in which the application of the new approach is presented and tested in a case study.

***Authors response:*** *As it was structured in the first submission, we intended to present the methodology for making Impact Webs as 'the result', as this is a methodological paper. This was discussed internally among the authors a number of times when drafting the publication, however, given reviewers comments, we agree with your suggestion and have restructured the paper as follows:*

**1. Introduction**
**2. Methodological development**
*2.1 Methodological pre-development: Scoping review of conceptual models of risk for inspiration*
*2.2 Selection of constitutive elements in an Impact Web*
*2.3 Steps for constructing an Impact Web*
*2.4 Trail in test cases*
**3. Results: Proof of concept**
*3.1 Complex risks linked to COVID-19, concurring hazards and responses in Guayaquil, Ecuador*
**4. Discussion**
*4.1 Strengths*
*4.2 Limitations*
*4.3 Future research direction*
**5. Conclusions**

Thank you for taking into account my remark and for implementing the advice.

13. The authors present in Section 3.1 the constitutive elements of an 'Impact Web'. Unfortunately, this is not very illustrative. It would be much better if the authors present these eight model elements in a separate table directly next to Figure 1. It would also be important for the recognition of the individual elements that the elements are not grouped together or named differently. For

example, the authors have combined "Interventions & response risks" and "Risks that did non manifest". I would strongly advise against this. It would also be good to use the same colors for the elements in such a table as in Figure 1.

*Authors response: Thank you for this suggestion. We have produced a new table next to Figure 1 which includes the elements, how we chose to visualize them, a short description and examples. We additionally agree to not group the elements, and have separated them.*

Thank you for taking into account my remark and for implementing the advice.

One more comment: to improve the readability, I would suggest using the exact same wording for the 'constitutive elements' in the text headings, in the table and in Figure 1. There are still discrepancies in the exact wording.

Text headings:
"Hazard, threats and shocks"
"Root causes of risk and vulnerability"

Figure 1 and Table2:
"Hazard, threats & shocks"
"Root causes of risk & vulnerability"

14. For the model element "Connections between elements", please explain the syntax in more detail. It is not entirely clear from the text what the different line types (solid, dashed) mean.

*Authors response: We have done this in the new table.*

Okay. Thank you.

15. For the model element "Multiple interacting hazards", please indicate that you use icons in the model.

*Authors response: We have done this.*

Okay. Thank you.

16. For the model element "Direct & cascading impacts", please explain that you distinguish between positive and negative impacts, which are indicated by cross and hook signs. Why did the authors choose these symbols? Wouldn't plus and minus signs be more intuitive?

*Authors response: We have done this in the table. We avoided the plus and minus signs to avoid confusion with the symbology in use in other modelling approaches (e.g. causal loop diagrams), where the causal connections between elements is indicative of positive or negative influence between each other. This is not the case with our model/ With impact Webs, we demonstrate the cascading effects between impacts, and what their drivers and root causes are, as well as what interventions were taken in response measures and how these create cascading effects themselves.*

Okay. Thank you.

17. The authors mention in L292 that during the scoping step an important step for selecting the scale to model by looking at geographical or administrative boundaries. It would be interesting to learn more about when the authors would suggest considering geographic or administrative boundaries. Risks can cross administrative boundaries. What would be the argument for looking at administrative boundaries anyway?

*Authors response: While we fully agree that risks cross administrative and geographic boundary's (We included multiple scales in the model for exactly this reason) there is always a difficulty when modelling complex/ systemic/*

*cascading risks, as there is a challenge of where to start and where to stop. We suggest to start with geographic and administrative boundaries to start with as it is practical, and helps to refine the context and objective of the assessment. This makes more sense to us that starting from a specific sector, as we want to model multiple hazards and their effects on multiple sectors. There is of course, no one correct way to go about it, and we reflect on this in the paper, both in the scoping section (Lines 397 -400) and in the discussion in the limitations section of Impact Webs.*

Okay.

18. L306-307: I don't get what you want to say with this sentence: "This perspective acknowledged that the systems relationships emerge more clearly when under stress, i.e. become more visible and therefore easier to observe." What do you mean with "under stress"? Please consider revising this sentence.

*Authors response: With under stress we mean that when impacts become observable in the system, it becomes easier to retrace the complex relationships that explain them and normally building up undetected in the background. Therefore, it made sense to start building from the observation of impacts. We will revise the sentence to be clearer.*

Okay. Thank you for the better explanation in the text.

19. In lines 364 to 367, the authors mention that they only select one case study to demonstrate the use of the new 'Impact Webs' approach. I fully understand why only one case study was selected. But why did the authors choose the Guayaquil case study and not one of the others? It would be good to provide a justification. Now, reading on, I see that the authors state in Section 3.3. "Step 1: Scoping" why they chose this case study. But perhaps it would be good to make this point earlier.

*Authors response: We have included a line to point to the justification of showing Guayaquil in the next section (Line 527-528).*

Okay. Thank you for mentioning this.

20. In lines 386 to 394, the authors summarize the results from step 2,3 and 4 for the Guayaquil case study. But actually, they only mention which elements are included in the final model. That provides no added value. The reader learns nothing about the process, how the elements were selected, how the workshops with the stakeholders went, whether there was disagreement, or how many elements there were initially and how many the model was reduced to in the end? More details about the actual steps 2, 3 and 4 would be extremely helpful for further future applications of the 'Impact Webs'. And also, how long did each step take? Please provide further details in this regard.

*Authors response: Section 3.3 in our first submission was a proof of concept from one of the test cases, we included it as we feel it's added value is demonstrating what an Impact Web and narrative storyline actual is in a final format. The process of how we got there is the previous steps in in sections 2 and 3 in submission 1, which we drafted in a manner that showed the overarching process across all of the cases. We have provided further details on the workshops and process in the 'Trail in test cases' section, as per your and the other reviewers feedback.*
*With regards to your specific questions, it was not our objective in this paper to elaborate in detail how the stakeholder engagement process went with the Guayaquil case (for example if specific stakeholders has disagreements on which model elements to select and how we facilitated such disagreements), as we do not think this adds much to the scientific discussion on the need for new complex risk assessment methodologies and how we developed a new conceptual model to do this. The objective of our paper is to demonstrate the overall methodological development and steps we followed to create a new methodology in Impact Webs. We intend for the publication to showcase Impact Webs, with Guayaquil as our 'proof of concept'*
*We have included more details on the stakeholder engagement in the new submission. However, some of the questions you ask for further details on are highly context specific, and elaborating on them would have required a different framing on the objective of the paper in the introduction, and on case study specifics when this is not a paper about Guayaquil specifically, thereby diverting from the focus of the paper, i.e. the development of the methodology for Impact webs.*

Thank you for the explanation. And thanks that nevertheless a few more details about the workshops in 2.3 have now been presented.

21. In "Step 5: Narrative storyline for Guayaquil, Ecuador", the authors provide the storyline they have developed for this case study. The story is very interesting and important for the later communication with stakeholders. But wouldn't it also be interesting for the reader to learn more about the process of writing this story? Who of the team wrote it? How was the story constructed? Was an overarching storyline followed? Were there difficulties? How often was the story reflected upon with the entire team? How long did the process take?

It would also be good to highlight the "actual story" in the text, for example by using italicized font. If I understand it correctly, the actual story goes from line 402 to 438. Or am I wrong?

*Authors response: We are glad to hear that you also think including this result is interesting and important for communication. As stated in response to comment 20, in this section we are demonstrating proof of concept, it is the 'result' of following the previous steps laid out in the article. We have further elaborated on how we derived the storyline in Step 5 of section 2.3, keeping in line with the overarching aims and objective of the article.*

Thank you for adding more details on how you derived the storyline in Section 2.3. And thanks for emphasizing the storyline using italics.

22. Figure 3: It would be important if the elements were listed in the same order in the legend as in Figure 1, which presented the basic structure of an 'Impact Web'. It would also be important to use the same terminology. In Figure 3, for example, the authors only refer to "Root cause" instead of "Root cause of risk and vulnerability".

*Authors response: We agree, and have done this.*

Thank you for taking into account my comment into consideration and implementing the advice.

23. In lines 439 to 451, the authors highlighted and described the advantages of the 'Impact Web' approach in the case study of Guayaquil. But that doesn't fit into "Step 5: Narrative storyline for Guayaquil, Ecuador" at all, does it? Wouldn't it be better in a separate paragraph? Somehow it is a bit misleadingly placed.

*Authors response: We have restructured the paragraph so it fits more closely to the narrative description.*

Okay.

24. In Section "4.1 Strengths", the authors say that the new approach "is useful to conceptualise, identify and visualise networks of interconnected elements across different systems and sectors." To prove this, it would be good to get a brief overview of the challenges and successes and lessons learned from the other case studies. A short overview table would be good for this. It might also be helpful to show the other impact web models of the other 4 case studies in the supplementary material.

*Authors response: The strengths and limitations sections in the discussion are summarizing reflections and lessons learned across all of the cases. We can introduce this at the start of the discussion, to elaborate that we are not just reflecting on our 'proof of concept' but from the entire research process that led us to these discussion points, particularly from our methodological development as well, given this is a methods paper. With the inclusion of further details of the other cases in the new section 2.4., we feel we can substantiate the argument.*

Thank you.

25. L472-473: "Given the models effectiveness for mapping the complexity of an event such as COVID-19 suggests that you could equally develop an Impact Web to understand the complexity of climate change risks." I don't understand this sentence. Is it related to what was said in the previous sentence, that a simple cause-effect chain model was developed first?

*Authors response: Here we are highlighting that Impact Webs were effective for modelling an event which was highly complex and cross-sectoral (COVID-19), and therefore, could likely be effective for modeling complex and cross sectoral climate change risks. We can restructure the sentence to be clearer.*

Thank you. It's clearer now.

26. In lines 482 to 484, the authors state that, apart from the outputs (the visual and the narrative storyline), it was rather the process of developing the model that stimulated critical reflection in the modeler and involved stakeholder. Unfortunately, however, we did not learn much about this process when the example of Ecuador was presented. Please elaborate on this in an appropriate place.

*Authors response: This is a reflection that we observed through our own process of developing the model and engaging stakeholders in workshops. We have elaborated on this further in the discussion (lines 658-661).*

Thank you for adding the additional information.

But in my opinion, one aspect has not been addressed well enough. I have read in many studies that 'collaborative modelling workshops have enhanced the stakeholder's understanding of complex risks/processes". I would be interested to know on what you base your statement. Did you ask them after the workshop (e.g., via a questionnaire) or are these just gut feelings? I would appreciate it if you could expand on that. Thank you.

**Technical corrections:**
There are some grammatical errors and typos. I have highlighted the ones I noticed, but I suggest that the authors read the manuscript carefully again.
1. L29: "the methodologies usefulness" ❼ "the methodologies' usefulness"
2. L87: "compared with Climate Impact Chains" ❼ "compared **to** Climate Impact Chains"
3. L116: "In section 2, we present our methodological development from Impact Webs." ❼ "In section 2, we present our methodological development **of** Impact Webs."
4. L118: Is the word "trailed" really appropriate in this context?
5. L179: There is a typo: "until we synthesized an agreed" ❼ "until we synthesized an**d** agreed"
6. L217: "Following the inclusion multiple hazards" ❼ "Following the inclusion **of** multiple hazards"
7. L267: There is a typo: "For exmaple" ❼ "For e**xa**mple"

*Authors response: Thank you for these technical corrections, we have implemented them.*

Thank you for the correction. However, some new points have arisen through the revision of the manuscript that require further technical corrections. Please address these as well. Thank you.

1. L93: "[..], we offer a new complex risk assessment methodology **in** Impact Webs […]" → The wording is a bit strange. Perhaps "in the form of" or "with" would be better than "in".
2. L98 and L478: "**made** Impact Webs" and "impact Webs were **made** in […]" → The term 'made' is perhaps not the most elegant way of describing the process. I suggest revising this wording.
3. L117: "[…] as follows: In section **two** we present [..] → You should not use numeral words when referring to the sections. Please be consistent with the use of numbers. In addition, there is a missing comma after "section 2".
4. L122: You need to mention that "in the results" is section 3. You mention section 2, section, 4 and section 5, but no section 3. This is inconsistent.

5. L141-142: "A non-systematic scoping review was chosen it has advantages for developing new methodologies." → Strange sentence structure. Isn't a semi-colon or punctuation mark missing here?
6. L148-149: "We provide **select** key references (see Table 1). → The wording is grammatically incorrect. Please rephrase.
7. L213: "[…] to create a model that **was** useful for; […]" → "is useful"?
8. L482: "[…] across different locations each with their own with unique challenges […]" → please remove the double "with"

As the manuscript still contains many typing and grammar errors, I recommend carefully re-reading the manuscript or having it proofread.

---

## Author Response (AR2)

**Review of egusphere-2024-2844: Comments 2nd Round**

Title: *Impact Webs: A novel conceptual modelling approach for characterising and assessing complex risks*

Authors: Edward Sparkes, Davide Cotti, Angel Valdiviezo Ajila, Saskia E. Werners, Michael Hagenlocher
* * *
**Reviewer 1 | Comments 2nd round**

*Recommendation: I recommend minor revisions.*

I thank the authors for revising their manuscript carefully following my major revisions recommended. The manuscript is almost ready for publishing. I have only some minor revision requests that concern my past comments.

For clarification: From now on (see below), all black texts are my comments from the first round, the red text are the authors' comments after the first round and I have now drawn my new comments for the second revision in blue. When referring to specific line numbers, I am referring to the tracked changes file the authors have resubmitted.

**Authors response:** *We would like to sincerely thank you for your constructive and thorough comments. Your review has sharpened the article, and we are glad of your support in shaping this contribution to the peer reviewed literature on complex risk assessment. I will provide my responses to your second round of comments in green italics.*
* * *
**Responses to reviewer 1**
**General comments:**

The article presents a novel conceptual complex risk assessment methodology named 'Impact Webs'. The authors describe the conceptualization of this new approach to complex risk modeling and how they developed it based on combining the advantages of several existing conceptual risk modelling approaches. The steps for constructing such a 'Impact Web' are also described, and at the end the authors show the results of a complex risk assessment using a case study in Guayaquil, Ecuador to provide proof of concept.

In general, the structure of the paper is rather unfortunate chosen or the headings are simply inappropriate. The authors describe in Section 2 "Methodological development" that they did a literature review and how this inspired the development and conception of their new 'Impact Webs'. Furthermore, they write that they carried out further theory and conceptual synthesis within the research team (but no further information is provided) to develop the concept for the impact webs. They also briefly mention that they tested the concept with stakeholders in five different case studies. And then in Section 2 "Results" the structure of these 'Impact Webs' and the method for constructing such 'Impact Webs' is described. I think that this Section 3 describes the actual method and not the results. And Section 2 is more of a chapter that describes the methodological pre-considerations. And in Section 3.3, an example is given of how such a development of an 'Impact Web' is carried out using a case study. For me, this is rather a "Results" Section, presenting the results of actually applying the new approach in a case study. Overall, I suggest reconstructing and renaming the sections 2 and 3 including the subsections.

While I find this new 'Impact Webs' approach very exciting and interesting and valuable, I think it has not been sufficiently presented by the authors. Some work needs to be done to make it possible to replicate the method in future studies. Please see my specific comments for more details.

There are also some typos in the manuscript. I would advise the author to go through everything carefully again.

Overall, I recommend major revisions.

*Authors response: We would like to express our thanks to the reviewer for their time to provide useful, constructive and detailed comments. We are glad to hear that they find the Impact Webs approach exciting, interesting and valuable. We agree with the feedback on the structure and have revised the overall structure of sections 2 and 3 of the paper as per your recommendations. Section 2.1 has become the 'Methodological pre- development: Scoping review of conceptual models of risk for inspiration', and we have significantly restructure the 'Lessons from the review' section based on your comments below. The 'Selection of constitutive elements' & 'Steps for constructing an Impact Web' have been made part of the methodology in sections 2.2 and 2.3 respectively, and we have provided more information and detail of the case studies in the 'Trail in test cases' section, moving it to the end of the methodology in 2.4. We have removed the section on concept development (see specific response below). Section 3.3, where we show our proof of concept, giving an example of how such a development of an 'Impact Web' was carried out using a case study is the new results section as per your recommendation. Please see the restructured outline below:*

**1. Introduction**
**2. Methodological development**
**2.1** *Methodological pre-development: Scoping review of conceptual models of risk for inspiration*
**2.2** *Selection of constitutive elements in an Impact Web*
**2.3** *Steps for constructing an Impact Web*
**2.4** *Trail in test cases*
**3. Results: Proof of concept**
**3.1** *Complex risks linked to COVID-19, concurring hazards and responses in Guayaquil, Ecuador*
**4. Discussion**
**4.1** *Strengths*
**4.2** *Limitations*
**4.3** *Future research direction*
**5. Conclusions**

*With this new structure, and additional details on the methodological development, the manuscript is clearer for replicating the Impact Webs method in future studies. We have responded to your specific comments below.* **When referring to edits made in response to your comments, we are referring to the tracked changes file we have resubmitted.**

Thank you for addressing my concerns about the structure of the paper. I really appreciate you revising the sections. It's much clearer now.

*Authors response: Thank you for this useful feedback*

**Specific comments:**

1. The authors mention the term 'response risks' in the abstract and later in several places in the manuscript. In lines 52 and 228, they mention this in connection with the literature. But this term is not used in this literature. What do the authors mean by this? Do they mean "human responses", "response options" or "response actions"? Or do they mean here "responses" as a driver of risk (risks due to human responses that not achieve the intended outcome etc.)? If they are introducing a new term, then it should be identified as such.
Otherwise, the use of direct quotation marks in line 228 is confusing, because, as stated, the term as used by the authors here is not mentioned in the cited literature.

*Authors response: We use response risks to refer to the risks arising from responses to risks and impacts (e.g responses not achieving their intended objectives, or having trade-offs). The concept and term were introduced in the paper by Simpson et al. (2021), then adopted and extensively used in the IPCC AR6 WGII report, Chapter 1 'Point of departure and key concepts' (e.g. Figure 1.5-part C). In that report, an explanation of response risk is provided in multiple instances, e.g. "The risks of climate change responses include the possibility of responses not achieving their intended objectives or having trade-offs or adverse side effects for other societal objectives", or "risks can arise from potential impacts of climate change as well as human responses to climate change." To address the comment, we have elaborated on the term to explain how it is defined in the literature in the introduction, following the IPCC point of departure and Simpson et al (2021) referencing (lines 53 – 54 in track change file), removed 'risks' from the abstract (line 15) and include reference to AR6 Chapter 1 (line 54). We have additionally removed the quotation marks on line 228 (now line 291) to avoid confusion.*

Thank you for clearing up the confusion.

In lines 32-33 the authors talk about "both positive and negative outcomes of disaster risk management practices". What do you mean by that? I suggest rephrasing it to better clarify what you mean by positive and negative outcomes.

*Authors response: This is referring to the methodology being useful to evaluate trade-offs in decision making. We elaborated on this point in the discussion, (lines 652 – 667). To make sure this is clear in the abstract, have update the line to read "The participatory process of developing Impact Webs promotes stakeholder engagement, uncovers critical elements at risk and helps to evaluate trade-offs in decision making by improving understanding of both positive and negative outcomes of disaster risk management practices." (lines 32 – 33)*

I am still not satisfied with the answer. I was concerned about the wording "positive and negative outcomes of disaster risk management practices". What do you mean by "positive" and "negative"? Co-benefits and trade-offs? Please rephrase the wording "positive" and "negative". What is positive and what is negative depends on one's point of view. A more elegant formulation would be desirable.

*Authors response: We have reformulated the line and removed the words 'positive and negative', which now reads:* "The participatory process of developing Impact Webs with stakeholders uncovers critical elements in systems at risk, and helps to evaluate co-benefits and trade-offs of decisions by uncovering how the outcomes of disaster risk management practices affect people, organisations and sectors differently."

2. In lines 80–82, the authors cite literature on other system mapping approaches on which they base their newly developed Impact Webs: "… such as Causal Loop Diagrams (Coletta et al., 2024), Fuzzy Cognitive Maps (Gómez Martín et al., 2020) and Bayesian Belief Networks (Scrieciu et al., 2021)." It would be good to provide further references per approach and to put "e.g." in front of them, since these are only mentioned as examples of many others.

*Authors response: Thank you for the suggestion, we have done this.*

Okay. Thank you for the revision.

3. When describing the methodological development of the Impact Webs, the authors mention on line 116 that they "show other conceptual risk modelling approaches" that they drew inspiration from, without mentioning, however, that this overview was based on a literature review. It would be helpful to make this clear at this point, since the authors also name the next subsection accordingly.

*Authors response: Thank you for the suggestion, we have done this.*

Thank you for the revision.

4. In lines 121 to 127, the authors explain their literature search conducted as part of the study. But the information given about the process is much too thin. The method is not sufficiently documented to allow a replication of the review. Which search string was used? In which database was the search conducted? Why was this database selected and not others? Was the search restricted to peer-reviewed literature? How many articles were screened in total? What are the exclusion and inclusion criteria? Why was no systematic literature search conducted?

*Authors response: While systematic literature reviews have many strengths and allow for replication, they are not pre-requisite for conceptual inspiration and for the creation of new methodological approaches. Adopting a non-systematic scoping review approach gave us exploratory flexibility, for example to include and discuss new publications as we came across them, as well as include aspects of methods that we had used and were familiar with in our past research experience. While scoping reviews are not replicable, they are very helpful for developing initial conceptual frameworks and for concept development general.*
*To avoid confusion around this stipulation, we have renamed the section "Methodological pre-development: Scoping review of conceptual risk models for inspiration" to emphasize that our review was a scoping review to support evidence synthesis and inspire concept development. We now also elaborated on why a scoping non-systematic review was chosen in the text below (Lines 137 – 149). We feel that the table presents a useful synthesis of conceptual models used in risk assessments, and has merit in being published as it can be useful for other researchers using these methodological approaches in their work.*

Thank you for clearing this up.

5. In Table 1, the authors provide an overview of conceptual risk modelling approaches. I find this table very helpful. I am just wondering what the different approaches are categorized by? The authors start with "Climate Impact Chains" and end with "Participatory System Mapping". It might make sense to categorize the approaches by the degree of integration of quantitative data

or the approaches' ability to capture the complexity of the system (from linear to non-linear approaches. Furthermore, I wonder what the key references in the last column are sorted by? Relevance? Does it matter who is listed first? I would suggest sorting them by year of publication.

*Authors response: We are glad that Table 1 is found helpful. The different approaches are categorized by how the authors in the selected key references named them. If an author stated in their paper the method they used was Climate Impact Chains or Fuzzy Cognitive Maps for example, we follow their terminology.*
*While we appreciate there can be other ways to categorize the approaches, we do not think it makes sense to overly complicate this categorization in Table 1, for example, by degree of integration of quantitative data or the approaches ability to capture complexity of the system. We highlight in the table where approaches have strengths for integrating quantitative data, and the columns 'Strengths' and 'Weaknesses' in a complex risk context provide information on the approaches ability to capture complexity of the system. While we do highlight methodological ambiguity and cross-over in approaches (lines 200-211), we feel that such a table in a simpler format, categorized by approach name, has value. We propose not to categorize by degree of integration of quantitative data, but will have organized the references by year of publication.*

Okay, I see your argument for not categorizing the approaches. But you haven't addressed the last part of my point 6: "[...] I wonder what the key references in the last column are sorted by? Relevance? Does it matter who is listed first? I would suggest sorting them by year of publication." I am only referring to the **"Key references"** in column 5 of your Table 1. I would sort them in descending order, starting with the most recent publication. For example, for the approach 'Climate Impact Chains':
Sett et al (2024)
Petutschni et al (2023)
Zebisch et al (2023)
Harris et al (2022)
Hagenlocher et al (2018)

*Authors response: We agree with your suggestion and have organized the reference list by date as you suggest.*

6. Furthermore, I doubt that the last category presented by the authors "Participatory System Mapping" is not really a stand-alone conceptual model-risk model development approach. Participatory systems modelling can be done using various systems mapping methods (e.g. FCM or causal loop diagrams) to formalize knowledge. The mapping of conceptual systems can be done both participatory or on the basis of literature research. "Participatory" merely describes the process and the way in which the knowledge is generated. All of the first five approaches could be carried out in a participatory way. See, for example, the studies:

• Videira et al. 2009. Scoping river basin management issues with participatory modelling: The Baixo Guadiana experience. https://doi.org/10.1016/j.ecolecon.2008.11.008

• Sahin et al. 2020. Developing a Preliminary Causal Loop Diagram for Understanding the Wicked Complexity of the COVID-19 Pandemic. https://doi.org/10.3390/systems8020020

• Melles et al. (2021). COVID 19: Causal Loop Diagramming (CLD) of Social-Ecological Interactions for Teaching Sustainable Development. In: Leal Filho, W. (eds) COVID-19: Paving the Way for a More Sustainable World. World Sustainability Series. Springer, Cham. https://doi.org/10.1007/978-3-030-69284-1_16

• Olazabal et al. 2018. Transparency and Reproducibility in Participatory Systems Modelling: the Case of Fuzzy Cognitive Mapping. https://doi.org/10.1002/sres.2519

*Authors response: Thank you for this observation. We agree and have removed the participatory system mapping category from the table. We have incorporated some of the strengths and weaknesses of participatory system mapping approaches in other categories where their methodological application was relevant, as we feel they are important aspects to draw attention to in conceptual risk modelling approaches, and are equally relevant for these other types of approaches.*

Thank you for taking into account my remark and for implementing the advice.

7. The text in lines 131-173 in subsection 2.1 on the "Lessons from the review" does not really fit with the results presented in Table 1. The authors introduce three broad types of approaches following a study by Elsawah et al. (2017) in this section. However, the connection to the categories

the authors have identified from their review (presented in Table1) is not entirely clear. I do not see the benefit of introducing this broad classification by Elsawah et al. (2017). The examples in the table are not organized according to this classification. It just seems so disconnected. It would be better if the authors simply focus on the lessons learned from their own review and discuss and compare the strengths and weaknesses of the identified methods, in order to derive what they used for their new approach. The authors do this from line 161 onwards, but it would be interesting if the authors elaborate on this further. I would advise the authors to shorten the section (L132-173) and focus more on the actual results of their review.

*Authors response: We agree, and have completely restructured this section based on your and other reviewers' feedback. Following your suggestion, we have highlighted the different strengths in the approaches presented in table 1 and discuss why we drew on these for Impact Webs. This has made the section more informative of which aspects of different methods we were inspired by, and more logically link the section to the information in Table 1.*

Okay. Thank you for taking into account my advice and revising the entire sub-section.

I would like to draw your attention to the new study by Hanf et al. 2025 on using causal loop diagramming to derive an integrated, holistic understanding of the complex urban flood risk & climate change adaptation nexus in the coupled socio-ecological system of the city of Hamburg. In addition to the system elements, they have also integrated interventions into their conceptual risk model in order to understand vicious cycles of systemic feedback loops that reinforce flood risk and hinder climate change adaptation processes. It also important to mention that they use a narrative-based approach to communicate the results. By linking the individual feedback stories, they derived an overall system narrative to make the identified feedback structure more concrete and to offer an invitation to an integrated discussion and debate. Perhaps you consider this study worthy of mention.

*Hanf et al. (2025). Towards a socio-ecological system understanding of urban flood risk and barriers to climate change adaptation using causal loop diagrams. International Journal of Urban Sustainable Development, 17(1), pp. 69–102. doi: 10.1080/19463138.2025.2474399.*

*Authors response: Thank you for drawing our attention to this insightful work. We have included reference to it in our article in a number of places where relevant, as it is an important contribution to the literature and our impact webs approach draws similarities with it.*

8. In line 172 to 173, I do not understand what the authors mean with "of what stakeholder value and want to protect." I am not sure what the word "protect" means in this context. Please consider revising the sentence.

*Authors response: We have revised these sentences to be clearer and provide more of an elaboration of what we mean here (lines 237-242).*

Thank you for taking into account my remark and for implementing the advice.

9.   In section 2.3, the authors describe the next steps in how they made this methodology feasible for practical application. They briefly state that they tested the methodology with groups of stakeholders in various case studies. What I am missing here is a better overview of the five different case studies. Simply saying that you have five different case studies and giving their names is not enough. Why were five case studies selected? Why these in particular? What characterizes each case study? A table would be nice that provides an overview of the uniqueness and challenges (including the complex risks) of each case study in comparison to the others.

What did the authors learn from each case study?
The authors further describe that there were two workshop rounds with a number of different stakeholders. Were there two workshop rounds in total or two workshop rounds for each case study? This is not entirely clear.

*Authors response: We have updated this section to give more details of the five case studies, including why they were selected, their characteristics and unique challenges. We have also provided more details on the*

*workshops. As we have restructured the paper, the workshop details are provided in section 2.3. We feel this is the most appropriate place to elaborate on them, given it is where we detail the steps we followed to make Impact Webs*

Thank you for addressing this and for providing a more comprehensive overview on the case studies.

10. In line 179, the authors mention that the identified various theories and concepts were synthesized within a "research team". However, the authors do not provide any further details about this research team. It would be useful to get a clearer sense of how this team and this synthesis process was organized. How big was the team? Did all belong to the same institution? What was the team composition? What disciplines were represented in the team? What about the disciplinary bias of the team? Were there members of this team who led this conceptual development? What was the expertise of these members? And how long did the process of this conceptual development take?

*Authors response: We feel it is best to remove this section as it will make the methodology clearer based on the new structure suggested by the reviewers. While developing Impact Webs, we held a number of internal conceptual development discussions within our team (the Vulnerability Assessment, Risk Management and Adaptive Planning section at UNU-EHS) where we brainstormed and critiqued ideas from one another. While this was done to inspire concept development and is part of any research work, the team and synthesis process was not systematically organized. Under the new structure of the paper, where we present the selection of elements and steps for construction in the methodology, rather than as results, we do not feel this section adds much to the reader.*

Okay.

But in L96 you still speak of a "research team". Either rephrase it without using 'research team' or I suggest adding a sentence that explains who the team is and how it is composed. Thank you.

*Authors response: We have removed reference to the 'research team' and clarified that we are talking about the publication authors.*

11. As already described in my general comments, I suggest renaming Section 2 and restructuring and renaming Section 3. The current structure is misleading. The way I understand the text, Section 2 contains only preliminary methodical considerations (for example, a preparatory literature search), while Section 3.1-3.2 describe the actual new method/approach of 'Impact Webs'. In fact, I see section 3.3 as a "Results" Section in which the application of the new approach is presented and tested in a case study.

*Authors response: As it was structured in the first submission, we intended to present the methodology for making Impact Webs as 'the result', as this is a methodological paper. This was discussed internally among the authors a number of times when drafting the publication, however, given reviewers comments, we agree with your suggestion and have restructured the paper as follows:*

**1. Introduction**
**2. Methodological development**
*2.1 Methodological pre-development: Scoping review of conceptual models of risk for inspiration*
*2.2 Selection of constitutive elements in an Impact Web*
*2.3 Steps for constructing an Impact Web*
*2.4 Trail in test cases*
**3. Results: Proof of concept**
*3.1 Complex risks linked to COVID-19, concurring hazards and responses in Guayaquil, Ecuador*
**4. Discussion**
*4.1 Strengths*
*4.2 Limitations*
*4.3 Future research direction*
**5. Conclusions**

Thank you for taking into account my remark and for implementing the advice.

The authors present in Section 3.1 the constitutive elements of an 'Impact Web'. Unfortunately, this is not very illustrative. It would be much better if the authors present these eight model elements in a separate table directly next to Figure 1. It would also be important for the recognition of the individual

elements that the elements are not grouped together or named differently. For example, the authors have combined "Interventions & response risks" and "Risks that did non manifest". I would strongly advise against this. It would also be good to use the same colors for the elements in such a table as in Figure 1.

*Authors response: Thank you for this suggestion. We have produced a new table next to Figure 1 which includes the elements, how we chose to visualize them, a short description and examples. We additionally agree to not group the elements, and have separated them.*

Thank you for taking into account my remark and for implementing the advice.

One more comment: to improve the readability, I would suggest using the exact same wording for the 'constitutive elements' in the text headings, in the table and in Figure 1. There are still discrepancies in the exact wording.

Text headings:
"Hazard, threats and shocks"
"Root causes of risk and vulnerability"

Figure 1 and Table2:
"Hazard, threats & shocks"
"Root causes of risk & vulnerability"

*Authors response: We have done this, thank you for the observation.*

13. For the model element "Connections between elements", please explain the syntax in more detail. It is not entirely clear from the text what the different line types (solid, dashed) mean.

*Authors response: We have done this in the new table.*

Okay. Thank you.

14. For the model element "Multiple interacting hazards", please indicate that you use icons in the model.

*Authors response: We have done this.*

Okay. Thank you.

15. For the model element "Direct & cascading impacts", please explain that you distinguish between positive and negative impacts, which are indicated by cross and hook signs. Why did the authors choose these symbols? Wouldn't plus and minus signs be more intuitive?

*Authors response: We have done this in the table. We avoided the plus and minus signs to avoid confusion with the symbology in use in other modelling approaches (e.g. causal loop diagrams), where the causal connections between elements is indicative of positive or negative influence between each other. This is not the case with our model/ With impact Webs, we demonstrate the cascading effects between impacts, and what their drivers and root causes are, as well as what interventions were taken in response measures and how these create cascading effects themselves.*

Okay. Thank you.

16. The authors mention in L292 that during the scoping step an important step for selecting the scale to model by looking at geographical or administrative boundaries. It would be interesting to learn more about when the authors would suggest considering geographic or administrative boundaries. Risks can cross administrative boundaries. What would be the argument for looking at administrative boundaries anyway?

*Authors response: While we fully agree that risks cross administrative and geographic boundary's (We included multiple scales in the model for exactly this reason) there is always a difficulty when modelling complex/ systemic/*

*cascading risks, as there is a challenge of where to start and where to stop. We suggest to start with geographic and administrative boundaries to start with as it is practical, and helps to refine the context and objective of the assessment. This makes more sense to us that starting from a specific sector, as we want to model multiple hazards and their effects on multiple sectors. There is of course, no one correct way to go about it, and we reflect on this in the paper, both in the scoping section (Lines 397 -400) and in the discussion in the limitations section of Impact Webs.*

Okay.

17. L306-307: I don't get what you want to say with this sentence: "This perspective acknowledged that the systems relationships emerge more clearly when under stress, i.e. become more visible and therefore easier to observe." What do you mean with "under stress"? Please consider revising this sentence.

*Authors response: With under stress we mean that when impacts become observable in the system, it becomes easier to retrace the complex relationships that explain them and normally building up undetected in the background. Therefore, it made sense to start building from the observation of impacts. We will revise the sentence to be clearer.*

Okay. Thank you for the better explanation in the text.

18. In lines 364 to 367, the authors mention that they only select one case study to demonstrate the use of the new 'Impact Webs' approach. I fully understand why only one case study was selected. But why did the authors choose the Guayaquil case study and not one of the others? It would be good to provide a justification. Now, reading on, I see that the authors state in Section 3.3. "Step 1: Scoping" why they chose this case study. But perhaps it would be good to make this point earlier.

*Authors response: We have included a line to point to the justification of showing Guayaquil in the next section (Line 527-528).*

Okay. Thank you for mentioning this.

19. In lines 386 to 394, the authors summarize the results from step 2,3 and 4 for the Guayaquil case study. But actually, they only mention which elements are included in the final model. That provides no added value. The reader learns nothing about the process, how the elements were selected, how the workshops with the stakeholders went, whether there was disagreement, or how many elements there were initially and how many the model was reduced to in the end? More details about the actual steps 2, 3 and 4 would be extremely helpful for further future applications of the 'Impact Webs'. And also, how long did each step take? Please provide further details in this regard.

*Authors response: Section 3.3 in our first submission was a proof of concept from one of the test cases, we included it as we feel it's added value is demonstrating what an Impact Web and narrative storyline actual is in a final format. The process of how we got there is the previous steps in in sections 2 and 3 in submission 1, which we drafted in a manner that showed the overarching process across all of the cases. We have provided further details on the workshops and process in the 'Trail in test cases' section, as per your and the other reviewers feedback.*
*With regards to your specific questions, it was not our objective in this paper to elaborate in detail how the stakeholder engagement process went with the Guayaquil case (for example if specific stakeholders has disagreements on which model elements to select and how we facilitated such disagreements), as we do not think this adds much to the scientific discussion on the need for new complex risk assessment methodologies and how we developed a new conceptual model to do this. The objective of our paper is to demonstrate the overall methodological development and steps we followed to create a new methodology in Impact Webs. We intend for the publication to showcase Impact Webs, with Guayaquil as our 'proof of concept'*
*We have included more details on the stakeholder engagement in the new submission. However, some of the questions you ask for further details on are highly context specific, and elaborating on them would have required a different framing on the objective of the paper in the introduction, and on case study specifics when this is not a paper about Guayaquil specifically, thereby diverting from the focus of the paper, i.e. the development of the methodology for Impact webs.*

Thank you for the explanation. And thanks that nevertheless a few more details about the workshops in 2.3 have now been presented.

20. In "Step 5: Narrative storyline for Guayaquil, Ecuador", the authors provide the storyline they

have developed for this case study. The story is very interesting and important for the later communication with stakeholders. But wouldn't it also be interesting for the reader to learn more about the process of writing this story? Who of the team wrote it? How was the story constructed? Was an overarching storyline followed? Were there difficulties? How often was the story reflected upon with the entire team? How long did the process take?

It would also be good to highlight the "actual story" in the text, for example by using italicized font. If I understand it correctly, the actual story goes from line 402 to 438. Or am I wrong?

*Authors response: We are glad to hear that you also think including this result is interesting and important for communication. As stated in response to comment 20, in this section we are demonstrating proof of concept, it is the 'result' of following the previous steps laid out in the article. We have further elaborated on how we derived the storyline in Step 5 of section 2.3, keeping in line with the overarching aims and objective of the article.*

Thank you for adding more details on how you derived the storyline in Section 2.3. And thanks for emphasizing the storyline using italics.

21. Figure 3: It would be important if the elements were listed in the same order in the legend as in Figure 1, which presented the basic structure of an 'Impact Web'. It would also be important to use the same terminology. In Figure 3, for example, the authors only refer to "Root cause" instead of "Root cause of risk and vulnerability".

*Authors response: We agree, and have done this.*

Thank you for taking into account my comment into consideration and implementing the advice.

22. In lines 439 to 451, the authors highlighted and described the advantages of the 'Impact Web' approach in the case study of Guayaquil. But that doesn't fit into "Step 5: Narrative storyline for Guayaquil, Ecuador" at all, does it? Wouldn't it be better in a separate paragraph? Somehow it is a bit misleadingly placed.

*Authors response: We have restructured the paragraph so it fits more closely to the narrative description.*

Okay.

23. In Section "4.1 Strengths", the authors say that the new approach "is useful to conceptualise, identify and visualise networks of interconnected elements across different systems and sectors." To prove this, it would be good to get a brief overview of the challenges and successes and lessons learned from the other case studies. A short overview table would be good for this. It might also be helpful to show the other impact web models of the other 4 case studies in the supplementary material.

*Authors response: The strengths and limitations sections in the discussion are summarizing reflections and lessons learned across all of the cases. We can introduce this at the start of the discussion, to elaborate that we are not just reflecting on our 'proof of concept' but from the entire research process that led us to these discussion points, particularly from our methodological development as well, given this is a methods paper. With the inclusion of further details of the other cases in the new section 2.4., we feel we can substantiate the argument.*

Thank you.

24. L472-473: "Given the models effectiveness for mapping the complexity of an event such as COVID-19 suggests that you could equally develop an Impact Web to understand the complexity of climate change risks." I don't understand this sentence. Is it related to what was said in the previous sentence, that a simple cause-effect chain model was developed first?

*Authors response: Here we are highlighting that Impact Webs were effective for modelling an event which was highly complex and cross-sectoral (COVID-19), and therefore, could likely be effective for modeling complex and cross sectoral climate change risks. We can restructure the sentence to be clearer.*

Thank you. It's clearer now.

25. In lines 482 to 484, the authors state that, apart from the outputs (the visual and the narrative storyline), it was rather the process of developing the model that stimulated critical reflection in the modeler and involved stakeholder. Unfortunately, however, we did not learn much about this process when the example of Ecuador was presented. Please elaborate on this in an appropriate place.

*Authors response: This is a reflection that we observed through our own process of developing the model and engaging stakeholders in workshops. We have elaborated on this further in the discussion (lines 658-661).*

Thank you for adding the additional information.

But in my opinion, one aspect has not been addressed well enough. I have read in many studies that 'collaborative modelling workshops have enhanced the stakeholder's understanding of complex risks/processes". I would be interested to know on what you base your statement. Did you ask them after the workshop (e.g., via a questionnaire) or are these just gut feelings? I would appreciate it if you could expand on that. Thank you.

*Authors response: We have expanded further in this paragraph to be clear that this learning was from receiving verbal feedback from workshop participants, the paragraph now reads: "While the final visual and the narrative storyline is the output, it is the process of developing an Impact Web that stimulates critical reflection in the modeler and involved stakeholders, which is one of the key outcomes. In each of the test cases, many of the stakeholders involved in the workshops entered with expertise in one specific sector or to share their own lived experience. However, many participants gave verbal feedback that after collaboratively working on the model together, they had learned from other and now better understood impacts and drivers outside of their area of expertise, and thus had a better understanding of complex and cross-sectoral risks."*

**Technical corrections:**
There are some grammatical errors and typos. I have highlighted the ones I noticed, but I suggest that the authors read the manuscript carefully again.
1. L29: "the methodologies usefulness" ❼ "the methodologies**'** usefulness"
2. L87: "compared with Climate Impact Chains" ❼ "compared **to** Climate Impact Chains"
3. L116: "In section 2, we present our methodological development from Impact Webs." ❼ "In section 2, we present our methodological development **of** Impact Webs."
4. L118: Is the word "trailed" really appropriate in this context?
5. L179: There is a typo: "until we synthesized an agreed" ❼ "until we synthesized an**d** agreed"
6. L217: "Following the inclusion multiple hazards" ❼ "Following the inclusion **of** multiple hazards"
7. L267: There is a typo: "For exmaple" ❼ "For e**xa**mple"

*Authors response: Thank you for these technical corrections, we have implemented them.*

Thank you for the correction. However, some new points have arisen through the revision of the manuscript that require further technical corrections. Please address these as well. Thank you.

1. L93: "[..], we offer a new complex risk assessment methodology **in** Impact Webs [...]" ❼ The wording is a bit strange. Perhaps "in the form of" or "with" would be better than "in".
   *Done*
2. L98 and L478: "**made** Impact Webs" and "impact Webs were **made** in [...]" ❼ The term 'made' is perhaps not the most elegant way of describing the process. I suggest revising this wording.
   *Done*
3. L117: "[...] as follows: In section **two** we present [..] ❼ You should not use numeral words

when referring to the sections. Please be consistent with the use of numbers. In addition, there is a missing comma after "section 2".

*Done*

4. L122: You need to mention that "in the results" is section 3. You mention section 2, section, 4 and section 5, but no section 3. This is inconsistent.

*Done*

5. L141-142: "A non-systematic scoping review was chosen it has advantages for developing new methodologies." ⑦ Strange sentence structure. Isn't a semi-colon or punctuation mark missing here?

*We have revised the sentence*

6. L148-149: "We provide **select** key references (see Table 1). ⑦ The wording is grammatically incorrect. Please rephrase.

*Done*

7. L213: "[…] to create a model that **was** useful for; […]" ⑦ "is useful"?

*Done*

8. L482: "[…] across different locations each with their own with unique challenges […]" ⑦ please remove the double "with"

*Done*

As the manuscript still contains many typing and grammar errors, I recommend carefully re-reading the manuscript or having it proofread.

***Authors response:*** *Thank you for your thorough proof of the manuscript. We have gone through and done a careful proof of the entire manuscript.*

**Reviewer 2 | Comments 2ⁿᵈ round**

The authors did a great job in responding to the previous round of comments, significantly improving the quality of the manuscript. I have no major comments and want to congratulate the authors to a nice and inspiring work. Below, I have added three minor comments for consideration.

*Authors response: We would like to thank you very much for your useful and constructive comments to the manuscript. The review has sharpened the article, and we are glad of your support in shaping this contribution to the peer reviewed literature on complex risk assessment.*

- L.162-203: I would suggest to use this as a start of the next section. It is not part of the scoping but already provides information about the intention of impact webs.

  *Authors response: We agree, we have done this*

- L.399-437: I would suggest to add the introduction of the case studies before explaining the steps you conducted for the development of the case studies. The reason is that you refer to the case studies before properly introducing them (e.g. l.381, l.370).

  *Authors response: We agree, we have done this*

- L. 569: ". . "
  *Done*
* * *
**Responses to reviewer 2 | Comments 1ˢᵗ round**

**Reviewer 2**

In the manuscript "Impact Webs: A novel conceptual modelling approach for characterising and assessing complex risks", the authors make use of expert judgment and non-systematic literature review to build on lessons from different approaches to come up with a new conceptual modelling approach for systemic risk characterization. It is a relevant topic, however the contextualization and purpose of the paper are not clearly supported. I would thus recommend major revisions as outlined in the following.

*Authors response: Thank you for your useful comments on our draft publication. We are glad you think it is a relevant topic. We feel that, upon integrating reviewer feedbacks, it will add value to the peer reviewed literature and scientific discussion on complex risk assessments. **When referring to edits made in response to your comments, we are referring to the tracked changes file we have resubmitted.***

**The structure** of the manuscript is very unclear. It is unclear what is result, what is method, and most often, how certain conclusions were drawn. The authors should revisit their structure and narrative for this paper. It is not clear to me, what this paper offers to its readers. Is it the pure idea of impact webs (as an advancement of impact chains), is it the visualization, is it the guidance to build impact chains? In its current form, the authors seem to do everything a bit, but nothing sufficiently in depth. Re-structuring and clarifying the objective of this paper. Just outlining the process of developing the web (as mentioned in the introduction), seems to fail answering a specific research question.

*Authors response: Our aim with this publication is to offer a new methodology to improve understanding of complex risks and to show how we made it. We agree that this could be made clearer in the introduction, and have done so in a second submission by setting out the aim clearly. To demonstrate how we achieved this aim, in submission 1 we offered the methodology of Impact Webs and showed how we made it as 'the result' – however based on you're and the other reviewers feedback we will significantly restructure sections 2 and 3 of the paper as follows: 1) Introduction (with more focus on the aim), 2) Methodology, 2.1) Methodological pre-development: Scoping review of conceptual risk models for inspiration, 2.2) Selection of elements, 2.3) Steps for constructing an Impact Web, 2.4) Trail in test cases (with more*

*details of the cases), 3) Results: Proof of concept, 3.1) Complex risks linked to COVID-19, concurring hazards and responses in Guayaquil, Ecuador. We will then keep the discussion in the same structure, reflecting on the aim and research gap. Please see the restructured outline below:*

*1. Introduction*

*2. Methodological development*
   *2.1 Methodological pre-development: Scoping review of conceptual models of risk for inspiration*
   *2.2 Selection of constitutive elements in an Impact Web*
   *2.3 Steps for constructing an Impact Web*
   *2.4 Trail in test cases*

   *3 Results: Proof of concept*
   *3.1 Complex risks linked to COVID-19, concurring hazards and responses in Guayaquil, Ecuador*

   *4 Discussion*
   *4.1 Strengths*
   *4.2 Limitations*
   *4.3 Future research direction*

   *5 Conclusions*

*With this new structure, and additional details on the methodological development, the manuscript will be clearer offer to its readers insights on how we developed a new methodology in Impact Webs.*

**What is Impact Webs?** It would be very valuable if authors could clarify what they mean when they refer to impact webs as a conceptual modelling approach. Part of what the authors present seems to be tools (how to visualize), some analysis guidance (see Figure 2). Overarchingly, it would be beneficial, if the authors could make it more explicit, what the purpose of impact webs is. They refer to Bayesian Belief Networks and other modelling methodlogies, include participatory elements which are then refined/complemented through desk studies. Are impact webs meant to be complete and/or correct? Used by who?

*Authors response: With Impact Webs we developed a conceptual model that aim to improve understanding of complex risks in the system or location being modelled (see lines 73-92). The methodology is flexible and can be applied in the chosen system the modeler wants to investigate (see lines 647-651). We drew on inspiration from Climate Impact Chains, Bayesian Belief Networks and other conceptual models which are used in risk assessments, which we highlight in table 2 through a non-systematic scoping review. We have significantly restructured the section 'lessons from the scoping review' to show more clearly what aspects of other models inspired us and how we selected different aspects from them for Impact Webs (see lines 197-252). In the paper we also show how we made impact webs, as its our intention for this to be a methodological paper where readers can replicate our method for their own setting (see sections 2.2 and 2.3). We now outline this in the revised manuscript in introduction (lines 93-94). We have made more explicit in the introduction what we mean when we refer to Impact Webs, and have laid out more clearly the purpose of Impact Webs in the introduction.*

**Method section:** This paper seems to heavily rely on expert judgment - which makes it very difficult to reproduce and to offer evidence regarding the claims offered here. One key question I had when reading this section was why the authors limited their search for inspiration to the field of single-hazard risk assessment instead of learning from fields that address similar or different complex systems (e.g. integrated water management, agent based modelling, system dynamics research community). If I understand correctly, the authors propose this method based on iterations/refinements in 5 case studies. At least a short introduction of these cases would already offer insight regarding the complex risk context/dynamics Impact Webs has been developed upon I would recommend taking inspiration from studies that have developed methodological approaches or investigated how such approaches have been

developed (e.g. McMeekin, 2020) to extend the method section and add an additional section covering the approach development process.

*Authors response: Based on your and the other reviewers' feedback have significantly restructured the methods section (see above response to the structure). In our scoping review, we did not limit our search to single-hazard approaches. We reviewed and drew inspiration from various other conceptual modelling approaches that are used in risk assessments. Some of these approaches are applied more for single-hazard risk assessments, and some more in multi-hazard or multi-risk assessments. Agent-based modelling was not one of the approaches that we included in our review as it is used less often used in a risk assessment context than the other methods, and we wanted to draw inspiration from more methods that did not rely heavily on computational tools and quantitative data (i.g. Impact Chains and Causal Loop Diagrams). Nearly all of the papers we include in table 1 integrate system dynamics and take a systems perspective, which is stated in stated 2.1 (lines 242-243). The reason we did not explicitly draw on integrated water management is because it is not a conceptual modelling approach. We have removed the participatory system mapping row from the table based on the other reviewers' feedback. We have updated the section 2.1 to make clear that this review was not exhaustive and was a scoping review, justifying why we chose to do this.*

*We have also include more details on the 5 case studies in the new section 2.4. We have removed the current section 2.2. 'concept development' due to reviewer feedback as it will make the new methodology structure clearer. While developing Impact Webs, we held a number of internal conceptual development discussions within our where we brainstormed and critiqued ideas from one another. While this was done to inspire concept development, the team and synthesis process was not systematically organized. Under the new structure of the paper, where we present the Selection of elements (2.2), Steps for construction (2.3) and Trail in test cases (2.4) in the methodology, we feel the approach development process is sufficiently covered and can be replicated in future research.*

**Regarding the Impact web development process:** Table 1 looks like something that would be worth for the Appendix or could be used in a shortened version to support a discussion of the different methods in the context of complex risk elements to be addressed with Impact Webs (section 3.1?). It would also be interesting to learn, why authors refer to storyline approaches as one of the steps in the impact web development process but did not consider them in Table 1 for inspiration. Section 3.1 seems a mix of presenting the complex risk elements of interest and mentioning what elements from which approach were used to visualize. I would suggest to separate these two purposes and rather provide more justification regarding the choices regarding the visual elements, e.g. by referring visualization research that justifies the choices. I also want to point out that terminology in Figure 1 is inconsistent (and not referred to in the paper). 'driver of risk', 'hazard', 'vulnerability' are all concepts that overlap (at least partially) and thus do not offer clear guidance what visualization element should be used.

*Authors response: We have significantly restructure section 2.1. based on your and other reviewers' feedback, and have highlighted the different strengths in the approaches presented in table 1 and discuss why we drew on these for Impact Webs. This now better supports a discussion of the different aspects of approaches that are useful in a complex risk context that inspired us. We did not include storyline approaches in table 1 as they are not conceptual modelling approach. However, we observed in the majority of papers from the review that the visual output of the model was also accompanied by narrative-based methods used to explore and communicate their findings, therefore we followed this. However, we realize we did not state this in submission 1 and have now updated this section (lines 248 – 252). We have updated section 2.2 and provided more justification regarding which elements we chose and why, and have included a new table next to Figure 1 which presents the elements, description, visual representation in the model and gives examples. We have also updated Figure 1 to be consistent with terminology, and have not grouped the elements in the subheadings in section 2.2 but separated them.*

The development steps (section 3.2) is unclear whether they are an outcome of the paper or the method to derive it. It is presented as a method (with limited justification why it is done that way), but lack insights/guidance into how the complexity of systemic risk (and the corresponding visualization) can be managed.

*Authors response: We have restructured the paper so this is now in the methodology – it was our original intention with the paper to present the method (i.e. how to make impact webs) as the result, but have change this based on both reviewers feedback. In the discussion, we reflect and give strengths and weaknesses into how the complexity of systemic risk can be understood and managed through the corresponding visualization and development of impact webs. We feel this is the best situated place for such a reflection in the paper.*

McMeekin, N., Wu, O., Germeni, E., and Briggs, A. (2020). How methodological frameworks are being

developed: evidence from a scoping review. BMC Med. Res. Methodol. 20, 173. https://doi.org/10.1186/s12874-020-01061-4

**Authors response:** *Thank you for providing this reference. This will be useful inspiration for restructured manuscript.*

**Responses to community comment from Luliana Armas**

**Community comment from Luliana Armas**

This paper is part of the ongoing efforts to design new operational and conceptual models that more effectively track risk propagation when multiple co-occurring or cascading hazards are involved. As researchers actively engaged in this field, we are glad to see more models emerging, and we are committed to contributing with insights that may help polish models such as Impact Webs.

*Authors response: Thank you for your constructive comments and insights. We are glad to receive them and are also enthusiastic to see new methodologies emerging.* **When referring to edits made in response to your comments, we are referring to the tracked changes file we have resubmitted.**

Accordingly, we commend the authors on their work and highlight several key points to address during this review:

**Use of concepts from the existing literature without definition or attribution to original models**
Impact Webs include "risks, their underlying hazards, risk drivers, root causes, responses to risks, as well as direct and cascading impacts" (lines 22-23). Although these components are briefly described in section 3.1., they are not clearly defined (e.g., hazard, shock, impacts, risk drivers, root causes). Moreover, the models that introduced these concepts are overlooked – for instance, the root causes that come from the PAR and Access models (Blaikie et al., 1994; Wisner et al., 2004), which are not explicitly mentioned in the paper. Also, the rationale for choosing these specific components for inclusion in Impact Webs and why they do not adhere to more intuitive component names (such as the ones of Impact Chains) are not discussed.

*Authors response: We feel that section 3.1 in the original submission describes well the elements that were selected, including our rationale for selecting them. In our new submission, we have edited the section where we introduce which elements we selected, more clearly defining them. We have also included a new table which presents the elements, description, visual representation in the model and gives examples (see table 2). The models that introduce root causes of risks and vulnerability are not overlooked, and are referenced in the original manuscript (See original manuscript lines 243 – 248). In the new submission we have provided more text to clearly define these components and their conceptual backing (see lines 306 – 332). We have significantly restructured section 2.1 of the manuscript to elaborate on what aspects of different models we selected and justified why we have done this. We highlight in the manuscript that we were heavily inspired by Impact Chains in the introduction, but wanted to overcome one of the critiques in the literature of Impact Chains that they are not well oriented to model complex and systemic risks, thus we drew on other conceptual models that engage with system dynamics to do this.*

**Overlooking the use of Impact Chains to analyze multi-risk**
The manuscript presents only the preliminary applications of Impact Chains, omitting recent advances and uses. In its current form, the paper fails to bring the reader up to date in terms of the ways Impact Chains (as inspiration for Impact Webs) are currently employed in the literature. Prominent research projects in the field of DRR, such as Paratus, use Impact Chains to assess systemic multi-sectoral and multi-hazard risk using a wide range of scenarios (Cocuccioni et al., 2024; Hurliman et al., 2024).

*Authors response: In our overview in table 1, we detail some of strengths of Climate Impact Chains in a complex risk context, and include reference to recent publications that go beyond preliminary applications of Impact Chains (see Sett et al., 2024 & Zebisch et al., 2023). In the introduction we recall the main strengths of, as well as some of the main critiques of Climate Impact Chains that have been prominent since their conceptualization over a decade ago. While new applications of Climate Impact Chains and related methodologies are emerging to tackle complex risks, these approaches are still on the periphery. We are pleased to see these approaches gaining traction, such as the two conference proceedings and the Paratus project that you provide reference too.*

**Failure to address the similarities between Impact Webs and recent conceptual models (i.e., Enhanced**

**Impact Chains)**

The authors acknowledge that their literature review on conceptual risk models is incomplete (line 123). Nevertheless, this literature review omits the model with the highest similarity to Impact Webs: the Enhanced Impact Chains (hereafter EICs) developed by Albulescu and Armaș (2024) and published in this very same Special Issue of NHESS. This shortcoming is understandable, given that this conceptual model was published only a month before this manuscript's submission.

In light of the following arguments, we believe that the authors should 1) include EICs in the literature review on the models used for inspiration and 2) address the novelty of Impact Webs by comparing them to EICs:

1. The models share the purpose of analyzing the interactions between risk elements. The difference is that EICs look at this problem through the lens of vulnerability dynamics, whereas Impact Webs adopt a broader approach. In essence, both serve as co-development tools that account for the complexity of risk assessment, standing out in terms of their capacity to organize a diverse and consistent volume of information, visualize it, and, based on it, evaluate the strengths and weaknesses of disaster risk management across multiple systems, sectors, and at various scales.

2. Both models include similar elements under different terminologies. For example, the vulnerabilities in EICs are the risk drivers and root causes in Impact Webs, the adaptation options in EICs are the responses to risk, while impacts and hazards are the same in both models.

3. Both models zoom in on cause-and-effect relationships while employing feedback loops to illustrate dynamic interactions among risk elements. In this particular case, EICs introduce named and clearly defined connections between the elements (including positive feedback loops), while Impact Webs do not name or describe the types of connections established among the elements. We recommend addressing this ambiguity on the types of connections included in the model.

4. Both models are applied in case studies involving multi-hazards represented by the COVID-19 pandemic and co-occurring natural hazards. The results from the two case studies should be compared in the Discussion section, as these are the only two multi-hazard case studies (including the COVID-19 pandemic) that employ conceptual models focused on the dynamics of risk elements.

5. Both models integrate stakeholder perspectives. In our paper on EICs, the level of stakeholder input is limited, but the model can be fully developed based on these perspectives (as the paper explicitly states).

6. Both models adopt a cross-sectoral approach, demonstrating high flexibility and allowing for the cross-comparison of results across different geographic and socio-cultural settings.

*Authors response: Thank you for highlighting the publication and methodology that was recently developed by yourself and colleagues, including the novelty of Impact Webs and EIC's. We cannot include EICs in our scoping review of conceptual models that we looked at for inspiration as this new and recently published methodology was not reviewed by us for inspiration. However, we feel it is important to acknowledge such new methods and have included reference to this in the discussion in the second submission (see lines 721-725).*

**Ambiguous terminology**
"Response to risk" is a term marked by ambiguity. Risks arise as the convergence of hazard,

exposure, and vulnerability, and to respond to them would mean addressing all three components. However, disaster risk management typically focuses on mitigating the vulnerability and impacts of the hazard—not the hazard itself. Therefore, we recommend changing the term to one that avoids confusion.

**Authors response:** *Response risks are referring to the risks arising from responses to risks and impacts (e.g responses not achieving their intended objectives, or having trade-offs). This terminology is well recognized, as it was included in the IPCC AR6 Chapter 1 'Point of departure and key concepts' (see Figure 1.5 part C). To make sure this is clear, we have elaborated on the term to explain it in more detail in the introduction, following the IPCC point of departure and Simpson et al (2021), and included references to both (see line 53).*